# Implementation and evaluation of open boundary conditions for sea ice in a regional coupled ocean (ROMS) and sea ice (CICE) modelling system

Pedro Duarte[1], Jostein Brændshøi[2], Dmitry Shcherbin[1], Pauline Barras[1], Jon Albretsen[3], Yvonne Gusdal[2], Nicholas Szapiro[2], Andreas Martinsen[1], Annette Samuelsen[4], Keguang Wang[2], Jens Boldingh Debernard[2]

[1] Norwegian Polar Institute, Fram Centre, Tromsø, Norway
[2] The Norwegian Meteorological Institute, Oslo, Norway
[3] Institute of Marine Research, Box 1870 Nordnes, 5817 Bergen, Norway
[4] Nansen Environmental and Remote Sensing Centre, Bergen, Norway

*Correspondence to*: Pedro Duarte (pedro.duarte@npolar.no)

**Abstract**

The Los Alamos Sea Ice Model (CICE) is used by several Earth System Models where sea ice boundary conditions are not necessary, given their global scope. However, regional and local implementations of sea ice models require boundary conditions describing the time changes of the sea ice and snow being exchanged across the boundaries of the model domain. The physical detail of these boundary conditions regarding, for example, the usage of different sea ice thickness categories or the vertical resolution of thermodynamic properties, must be considered when matching them with the requirements of the sea ice model. Available satellite products do not include all required data. Therefore, the most straightforward way of getting sea ice boundary conditions is from a larger scale model. The main goal of our study is to describe and evaluate the implementation of time-varying sea ice boundaries in the CICE model using two regional coupled ocean-sea ice models, both covering a large part of the Barents Sea and areas around Svalbard: the Barents-2.5 km, implemented at the Norwegian Meteorological Institute (MET), and the S4K, implemented at the Norwegian Polar Institute (NPI). We use the TOPAZ4 model and a Pan-Arctic 4 km-resolution model (A4) model to generate the boundary conditions for the sea ice and the ocean. The Barents-2.5 km model is MET's main forecasting model for ocean state and sea ice in the Barents Sea. The S4K model covers a similar domain but it is used mainly for research purposes. Obtained results show significant improvements in the performance of the Barents-2.5 km model after the implementation of the time-varying boundary conditions. The performance of the S4K model in terms of sea ice and snow thickness is comparable to that of the TOPAZ4 system but with more accurate results regarding the oceanic component because of using ocean boundary conditions from the A4 model. The implementation of time-varying boundary conditions described in this study is similar regardless of the CICE versions used in different models. The main challenge remains the handling of data from larger models before its usage as boundary conditions for regional/local sea ice models,

since mismatches between available model products from the former and specific requirements of the latter are expected,
implying case-specific approaches and different assumptions. Ideally, model setups should be as similar as possible to allow a
smoother transition from larger to smaller domains.

## 1 Introduction

Global, Arctic or Antarctic wide applications of the CICE model do not require any specific treatment regarding sea ice
boundary conditions because the model domain is larger than the areas where sea ice may occur. However, this is not the case
of regional implementations of the CICE or any other sea ice models. For such regional cases the past and current versions of
CICE include a simple way of dealing with open boundaries, restoring them every time step to the initial ice state or to some
predefined value, using a relaxation time scale. In the words of Hunke et al. (2015), this implementation is only intended to
"provide the hooks" for more sophisticated treatments. Therefore, the main goal of our study is to describe and evaluate the
implementation of sea ice time-varying boundaries in the Los Alamos Sea Ice Model using two regional models: the Barents-
2.5 km, implemented at the Norwegian Meteorological Institute (MET), and the S4K, implemented at the Norwegian Polar
Institute (NPI). We have chosen to use these two models because the former is an operational forecasting system, using data
assimilation and used for relatively short-term simulations (a few days), the latter is a research tool used for hindcast and
forecast longer-term simulations (a few years), without data assimilation, and this allowed us to evaluate the time-varying
boundary scheme for different types of models and simulations.
The use of sea ice models developed for large scales (like CICE) for small scale forecasts was discussed by Hunke et al (2020).
On the scales of the Barents-2.5 km and S4K model, the use of a continuum hypothesis and the viscous plastic rheology is far
from optimal. However, for coupled sea ice - ocean forecasts, good thermal and dynamical forcing and handling of ice-ocean
fluxes are also very important for the usefulness and quality of the forecasts. Also, knowledge about the possibility of ice in
an area might be more important for applications, such as navigation, than the specific details of the sea ice cover. Therefore,
we think adding capability to handle open boundary conditions in the sea ice model can increase the usefulness of small scale
regional coupled model systems for many applications.
Examples of regional implementations of sea ice models may be found in e.g. Smedsrud et al. (2006), Rousset et al. (2015)
and Prakash et al. (2022). Smedsrud et al. (2006) used The Regional Ocean Modeling System (ROMS,
https://www.myroms.org/) to run a high-resolution model of a polynya within a larger domain model. ROMS was used both
for the ocean and the sea ice. A relaxation open boundary scheme was used for ocean and ice variables between the nested
models. No details are given about the implied technicalities. Prakash et al. (2022) forced sea ice variables in their regional
ocean domain and, therefore, did not need to impose sea ice boundaries of any type. Rousset et al. (2015) describe the

implementation of lateral boundary conditions in the Louvain-La-Neuve sea ice model LIM3.6. We are not aware of any comprehensive description of sea ice time-varying boundaries for the CICE model.

## 2 Methods

### 2.1 Model description

We use The Regional Ocean Modeling System (ROMS, https://www.myroms.org/) and the Los Alamos Sea Ice Model (CICE). The software changes described herein are focused on the latter model. The CICE model is managed by the CICE Consortium with an active forum (https://bb.cgd.ucar.edu/cesm/forums/cice-consortium.146/ and a git repository https://github.com/CICE-Consortium). It includes two independent packages: CICE and Icepack. Sea ice dynamics is handled by CICE and sea ice columnar processes (thermodynamics and biogeochemistry) are handled by Icepack. Previous versions did not have such a separation, but the code evolved over the last years towards a clear distinction between processes which are mainly horizontal and those that are mainly vertical/columnar (since CICE 6). Various (older) versions of the CICE model are still in use by several modeling systems, including some Earth System Models that are part of CMIP6 [e.g. CICE 4.1, 5.1 and 5.1.2, see Roberts et al. (2015), Rasmussen et al. (2018), Wei et al., (2020), Smith et al. (2021)]. Scientific and technical details about the Los Alamos Sea Ice Model may be found in Hunke et al. (2015), Jeffery et al. (2016) the forum, and the Git repository mentioned above.

### 2.1.1 Coupling between ROMS and CICE

The coupling between ROMS and CICE was implemented at the Norwegian Meteorological Institute using The Model Coupling Toolkit (MCT, https://www.mcs.anl.gov/research/projects/mct/) and creating the METROMS framework mentioned above (e.g. Fritzner et al., 2019, https://doi.org/10.5281/zenodo.5067164). An early version of METROMS was also used by Naughten et al. (2017; 2018) and the coupling was very briefly described in those papers. ROMS is the controlling software acting through the CICE drivers CICE_InitMod.F90, CICE_RunMod.F90 and CICE_FinalMod.F90 to initialize, run and finalize CICE [these drivers are called from ROMS master routine (master.F)]. The variables exchanged through MCT are detailed in Table 1. The underlying philosophy behind the coupling is that fluxes are calculated in the model with most details of the underlying process, and then passed conservatively to the other. Thus, all fluxes except the production of 'frazil ice' are calculated in the ice model. Frazil ice production is simplified. First, the energy used to increase ocean temperature to the freezing point is calculated in ROMS when forcing has produced under-cooled water. This energy deficit is then passed to the CICE model (frzmlt variable in Table 1) and converted to a suitable amount of consolidated ice with heat and salt content consistent with the forcing. Any salt expelled from the ice by this process is then passed back again to ROMS.

Exchange frequency between the models depends on synchronization timestep and must be a common multiple of involved model timesteps. In default setups the models run concurrently on separate sets of compute cores, with a delayed exchange of fields, such that information calculated in one component is used in the other at the next coupling time interval. The coupled variables are declared in both ROMS and CICE and transferred both ways through MCT routines utilizing the underlying MPI library.

**Table 1. Data exchange between ROMS and CICE through MCT (see text).**

| From ROMS to CICE | | From CICE to ROMS | |
| --- | --- | --- | --- |
| Name and abbreviation | Dimensions | Name and abbreviation | Dimensions |
| Sea surface salinity (sss) | psu | Ice concentration (aice) | dimensionless |
| Sea surface temperature (sst) | ˚C | Freshwater flux from ice (freshAI) | kg s$^{-1}$ |
| Melt-freeze potential (frzmlt) | W m$^{-2}$ | Salt flux from ice (fsaltAI) | kg s$^{-1}$ |
| Velocity components (u and v) | m s$^{-1}$ | Nonradiative heat flux from ice (fhocnAI) | W m$^{-2}$ |
| Free surface height (ssh) | m | Radiative heat flux through sea ice (fswthruAI) | W m$^{-2}$ |
| | | Stress components in x-direction and y-directions (strocnx and strocny) | N m$^{-2}$ |

**2.1.2 Barents-2.5 km model**

The Barents-2.5 km model is MET Norway's primary model for forecasting of sea ice conditions in the northern regions. It consists of a fully coupled ocean and sea ice model that covers the Barents Sea and areas around Svalbard (Fig. 1). The modelling system employs the METROMS (https://doi.org/10.5281/zenodo.5067164) framework which implements the coupling between the ocean component (Regional Ocean Modeling System, ROMS3.7, https://www.myroms.org/) and the sea ice component (The Los Alamos Sea Ice Model, CICE5.1.2, https://www.osti.gov/biblio/1364126-cice-los-alamos-sea-ice-model) (e.g. Fritzner et al., 2019) (for details on coupling refer to 2.1.1). The model uses a grid with equally spaced points (2.5 km) in the horizontal, and differentially spaced (42 layers) terrain-following vertical coordinates (as the standard ROMS). The ice is distributed among 5 thickness categories with the lower boundary values: 0.00, 0.64, 1.39, 2.47 and 4.57 m. There are

7 vertical layers and one snow layer for each category. Both the ocean and sea ice utilize atmospheric forcing by AROME-Arctic, MET Norway's own numerical weather prediction model for the Arctic (https://www.met.no/en/projects/The-weather-model-AROME-Arctic; Müller et al., 2017). Considering that this model uses the exact same spatial grid as Barents-2.5 km, our ocean and sea ice experience atmospheric forcing without the loss of accuracy through processes like e.g. interpolation. Both ocean and sea ice use boundary conditions from TOPAZ4 (Sakov et al., 2012; Xie, 2017), which is a well-tested and documented assimilative (ensemble Kalman filter) coupled ocean and sea ice model covering the Arctic and North Atlantic oceans with operational fields readily available daily. TPXO7.2 tidal model (Egbert & Erofeeva, 2002) is used for tidal input. The river runoff climatology is based on the Norwegian Water Resources and Energy Directorate (NVE, http://nve.no) data for mainland Norway (Beldring et al., 2003) and AHYPE hydrological model for Svalbard and Russia (https://www.smhi.se/en/research/research-departments/hydrology/hype-our-hydrological-model-1.7994). The bathymetry is a smoothed version made from the IBCAO v3 dataset (Jakobsson et al., 2012). Operationally, the model assimilates AMSR2 sea ice concentration from the University of Bremen (https://seaice.uni-bremen.de/data/amsr2/asi_daygrid_swath/n6250/) over a 24 hour analysis run (details on assimilation and downscaling are given below 2.3.1).Then, using the improved initial condition, a 66-hour forecast is produced. The operational archive of the model is located at https://thredds.met.no/thredds/fou-hi/barents25.html. In this model, ocean boundaries are open, whilst sea ice boundaries were closed, until the implementation of the time-varying boundaries described in this work. The model has been run operationally from March 2019 and its results were evaluated against observations.

**2.1.3 S4K model**

The S4K (the Svalbard 4km) model has a slightly different domain than the Barents-2.5 km model (Fig. 1) and lower horizontal (4 km) and vertical (35 sigma layers) resolution in the ocean, while the configuration of ice thickness categories and vertical discretization is the same in both setups. The domain covers a slightly different area to allow producing boundary conditions for fjord models in Eastern Greenland. It is based on METROMS coupled with an earlier "columnar" version of CICE [with a "column package" for thermodynamics and biogeochemical processes developed as part of the Accelerated Climate Model for Energy (ACME) project, close to CICE6.0.0 alpha (https://github.com/CICE-Consortium/CICE/wiki/CICE-Release-Table)] following the same procedure described above for the Barents-2.5 km model (https://doi.org/10.5281/zenodo.5815093) (cf. – 2.1.2). The ocean and sea ice are forced with atmospheric fields from ECMWF Reanalysis v5 (ERA5, https://www.ecmwf.int/en/forecasts/dataset/ecmwf-reanalysis-v5). River forcing is based on: ArcticRims (https://rims.unh.edu), for Russia and North America, catchment area discharge estimates from the NVE (http://nve.no) for Northern Norway, and Mernild and Liston (2012) for Greenland. Sea ice boundary conditions are from TOPAZ4 (Sakov et al., 2012; Xie, 2017) and ocean boundary conditions are from the A4 model (Hattermann et al., 2016). This model was run continuously from August 2014 until July 2015 and its results evaluated against observations detailed in 2.3.2.

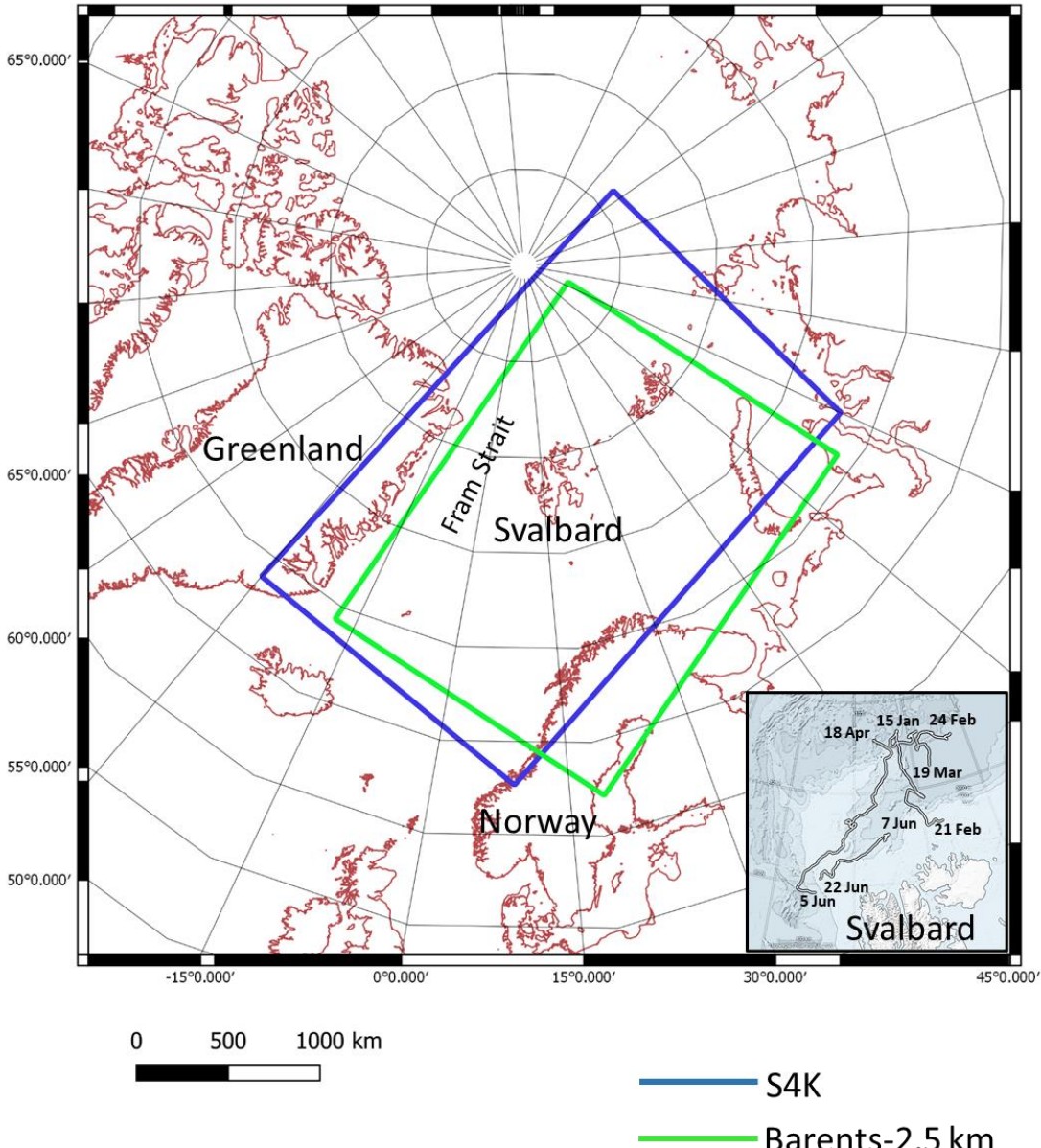

Figure 1. Barents-2.5 km and S4K model domains. The insert at the right bottom corner represents Svalbard and the area where the various drifts (lines showing the begin and end dates of each drift) of the N-ICE2015 expedition (Granskog et al., 2018) took place and along which sea ice and ocean data detailed in Table 3 were collected.

## 2.2 Implementation of time-varying boundary condition in the Los Alamos Sea Ice Model

### 2.2.1 Software details

We describe the main code changes in Table 2. We defined a Boolean variable (sea_ice_time_bry) that must be set to *True* in the CICE input file (ice_in) whenever time-dependent boundary fields are used. The main CICE model drivers CICE_InitMod.F90 and CICE_RunMod.F90 were modified. The first one initializes, and the second runs the model. The initialization driver now includes a call to a routine located in the file containing CICE forcing routines (ice_forcing.F90) that initializes boundary variables when sea_ice_time_bry = *True*. Similarly, the run driver includes a call to a subroutine in ice_forcing.F90 that updates the boundary variables at each time step. Updating implies reading boundary fields from boundary files and interpolating them to the model time step. Details on the boundary files are given below.

The new boundary variables match CICE variables. They have a prefix corresponding to the name of the corresponding variable in CICE (Table 2) followed by an underscore and the suffix "bry". We separated the new variables into ice-category-dependent two and three dimensional (2D and 3D) and ice-category-independent (Table 2). 2D variables represent either surface sea ice properties or bulk properties of ice or snow. 3D variables represent properties that vary vertically in the ice or snow and are resolved as a function of the number of ice and snow layers defined for a simulation. The ice-category-dependent variables have a dimension used to store the values of different ice thickness categories, defined as a function of sea ice thickness. For details on CICE size thickness categories see Hunke et al. (2015).

We allocate to the boundary variables the same dimensions allocated for the matching CICE variables, even though we need to track their values only along the open boundaries. This occupies more memory than necessary, with boundary variable "working" rectangular arrays being filled with zeros except for the boundary cells, but it simplifies the process of scattering variable values among different tiles in a parallel run, since we may reuse CICE data scattering routines. However, as described below, the boundary NetCDF files have only vector arrays and do not require "extra" space as the working arrays (see below).

The CICE file with more modifications for the time-varying boundary implementation is ice_forcing.F90 (Table 2). New routines were created to construct boundary file names, to read these files and to make the necessary time interpolations. Some specific file reading routines were implemented in ice_read_write.F90 given the format of boundary files (see below). These routines are called from ice_forcing.F90.

Boundary restoring takes place in file ice_restoring.F90, where the boundary values updated in ice_forcing.F90 are used to modify the corresponding CICE variables using a relaxation time defined in ice_in (trestore), along the "halo" cells (Hunke et al., 2015) located at the Northern, Southern, Western and Eastern limits of the model domain and their neighbor cells within the domain. These updates occur in the routine ice_HaloRestore that was modified from its original version. Snow and ice enthalpies are calculated from corresponding temperatures. In the tests carried out so far, we "relaxed" only the cells detailed above to follow exactly the way CICE deals with boundary conditions but a more complex treatment involving a larger relaxation zone may be considered.

**Table 2. Summary of main changes in the Los Alamos Sea Ice Model related with the implementation of time-varying boundaries**
(https://doi.org/10.5281/zenodo.5067164 **and** https://doi.org/10.5281/zenodo.5815093) **(see text).**

| Modified files | Main changes |
|---|---|
| ice_in | The Boolean sea_ice_time_bry was added to the domain name list. Time-varying boundary code is used when this variable is set to true. |
| CICE_InitMod.F90 | A call to init_forcing_bry - a new subroutine implemented in ice_forcing.F90 (see below) used to initialize the boundaries if the Boolean sea_ice_time_bry is set to true in the model input file (ice_in, see below). |
| CICE_RunMod.F90 | A call to get_forcing_bry - a new subroutine implemented in ice_forcing.F90 (see below) used to update the boundaries from corresponding files if the Boolean sea_ice_time_bry is set to true in the module input file (ice_in). |
| ice_forcing.F90 | **New variables were defined to store boundary values**. These parallel all model variables updated by the Los Alamos Sea Ice Model in ice_restoring.F90.<br>Ice-category dependent horizontal (2D) variables:<br>aicen_bry (ice concentration), vicen_bry, [ice volume per unit area (m)], vsnon_bry [snow volume per unit area (m)], alvln_bry (concentration of level ice), vlvln_bry [volume per unit of area of level ice (m)], apondn_bry, (melt pond fraction), hpondn_bry [melt pond depth category (m)], ipondn_bry [mean pond ice thickness (m)], Tsfc_bry [ice/snow surface temperature (°C)].<br>Ice-category dependent and vertically resolved (3D) variables:<br>Tinz_bry [sea-ice inner temperature (°C)], Sinz_bry (sea-ice inner bulk salinity) and Tsnz_bry [snow inner temperature (°C)].<br>Ice-category independent horizontal (2D) variables:<br>uvel_bry and vvel_bry [x (north/south) and y direction (west/east) velocity components (m s$^{-1}$)].<br>New routines were created:<br>init_forcing_bry - calculates current year and final year in forcing cycle.<br>boundary_files - constructs boundary file names from current simulated year.<br>boundary_files (and file_year_bry) - constructs boundary file names from current simulated year.<br>get_forcing_bry - calls boundary_data.<br>boundary_data – defines working arrays for boundary variables, call routines to read boundary files and to interpolate variable values to the model time step.<br>read_bry_ice_data_nc - this is an interface with the following procedures: read_bry_ice_data_nc_2D, read_bry_ice_data_nc_3D, read_bry_ice_data_nc_4D, to read boundary values from NetCDF files, according to their dimensions calling routines available in ice_read_write.F90 (see next Table line).<br>interpolate_data_n or interpolate_data_n_layer - interpolate boundary data between two consecutive time steps. The former and the latter are used for ice-category dependent 2D and 3D variables, respectively. Other variables reuse the "standard" interpolation routine (interpolate_data). |
| ice_read_write.F90 | Three routines (ice_read_nc_bry_2D, ice_read_nc_bry_3D and, ice_read_nc_bry_4D) were added to the interface ice_read_nc to read the different types of boundary data (see above). |

| | |
|---|---|
| ice_restoring.F90 | ice_HaloRestore - This is where boundary values are restored, using boundary data and a relaxation time scale (trestore) user-defined in the model input file (ice_in). |

Minor adjustments were implemented for Barents-2.5 km to enhance reliability for the operational system, particularly to blend

mismatches between the external and internal solutions. In ice_HaloRestore, the first physical points as well as the halos are

restored/nudged. Dynamical variables uvel, vvel, divu, shear, and strength are restored to the neighboring interior point.

Several technical additions address edge cases. Additional grid variables are extrapolated to halo cells (ice_grid.F90). Halo

cells are no longer zeroed during multiprocessor communications (ice_boundary.F90). Boundary values are restored before

both thermodynamics and dynamics (in CICE_RunMod.F90), which is necessary for prescribing boundary values (i.e., when

trestore=0).

In the S4K model, the only exception in the boundary restoring process is with uvel and vvel, which are restored as any other

boundary variable when there is sea ice outside the domain, else internal velocities are assumed in line with Rousset et al.

(2015). This is to guarantee that the sea ice motion inside the model domain is properly affected by larger scale drift trends in

"long-term" simulations (several months).

Our approach differs from that described by Rousset et al. (2015) for the lateral boundary conditions in The Louvain-La-Neuve

sea ice model LIM3.6 in that we restore tracer boundary values irrespective of the velocity direction across the boundaries.

Moreover, we do not fill the boundaries with ice thickness categories following a statistical law – categories are filled

depending on their availability in the available boundary data. In any case, specific changes can be easily made in the code to

test different settings.

**2.2.2 Boundary data details**

The main challenge with the boundary data is the matching between available model output for a larger domain and the data

needs of CICE. In the examples provided here we used data from TOPAZ4 as explained above. The available outputs relevant

for CICE boundaries include daily values for: ice concentration, ice and snow thickness, and ice east-west and south-north

velocities. There is no data for ice or snow internal or surface temperatures, or for ice salinity. There is no data of any kind of

ice thickness categories. Therefore, we had to make some assumptions. These will have to be defined for each application

depending on available boundary data. In our case we proceeded as follows:

1) TOPAZ values located along the boundaries of our domains were linearly interpolated to our grids.

2) Ice-category-dependent variables were stored in boundary files assuming the same number of categories used in our

runs (5). For each grid point, all values were set to zero, except for the category where available "bulk" ice thickness

belonged.

3) Surface (skin) snow or ice temperatures (in the absence of snow) were set to air temperatures taken from the atmospheric forcing files, when air temperature was < 0, else they were set to a slightly negative value (-0.00001 °C).

4) Inner snow and ice temperatures were obtained by linearly interpolating between the surface temperature and the freezing water temperature. The same temperature trend was assumed for snow and ice. Therefore, when snow was present its height was taken into account as the thickness of each ice layer.

5) Inner ice salinities were calculated to match multiyear and first year ice (MYI and FYI, respectively) profiles described in the literature (Gerland et al., 1999). We assumed that when ice thickness was > 1.5 m it was MYI, else it was FYI. In the case of MYI we used the profiles described in older versions of CICE (Hunke et al., 2015, equation 76). In the case of FYI we assumed a "C" shaped profile defined by equation 1 (e.g. Figure 3 of Gerland et al., 1999):

$$S_i = 19.539Z_i{}^2 - 19.93Z_i + 8.913 \qquad \text{(eq. 1)}$$

Where, $S_i$ is the salinity and $Z_i$ is the fractional depth of layer $i$ – zero at the ice top and 1 at the ice bottom.

Examples of boundary files may be found at: https://doi.org/10.5281/zenodo.5798076

## 2.3 Data used for model evaluation

### 2.3.1 Barents-2.5 km model

The data used to evaluate the Barents-2.5 km model can be found in Table 3. For this model system, the focus was purely on remote sensing of sea ice concentration. AMSR2 (https://seaice.uni-bremen.de/sea-ice-concentration/amsre-amsr2/) is a Passive Microwave product with a spatial resolution of 6.25 km (Spreen et al., 2008), consisting of continuous sea ice concentration values (SIC) between 0 and 1.0 (same as the model). The Norwegian ice charts (Dinessen & Hackett, 2016) have a gridding resolution of 1km and are produced manually based on multiple data sources, where the primary source is radar data (SAR). Since the ice charts consist of discrete values, the modeled SIC is categorized as shown in Table 4. For AMSR2, continuous values are applied. The satellite products are interpolated to the model resolution of 2.5 km, using bi-linear interpolation for the ice charts, and nearest neighbor method (same product as used for assimilation) for the AMSR2 products. In the comparison, all SIC > 0 are included, where land, missing values and open water (in both observations and model) are masked out. This means that the entire sea ice covered area inside the domain of the model is included in the comparison. The AMSR2 products are available daily, whereas the Norwegian ice charts, are only available during working days.

The data assimilation applied in the operational Barents-2.5 km model is the combined optimal interpolation and nudging (COIN; Wang et al., 2013). It was originally developed for assimilating sea ice concentration in a two-level sea ice model within ROMS and is now further developed for the multi-category CICE model in METROMS

(https://doi.org/10.5281/zenodo.5067164). The details of the method will be described in an upcoming paper (Wang et al., in prep.). The COIN method is a nudging method applied inside the CICE code. The modeled sea ice concentration is updated every model (CICE) time step with a small innovation (difference between model results and observations) such that the final analysis will reach the optimal estimate, which is a linear combination of the model results and the observations based on their variances (Wang et al., 2013). The daily AMSR2 sea ice concentration is assimilated, where the observations standard deviation is calculated according to Spreen et al. (2008), and the model standard deviation is approximated as the absolute difference between the model results and observations following Wang et al. (2013). During the assimilation, the real thickness of each category of snow and sea ice remains unchanged, so their volumes are updated according to the change of the ice concentrations.

**Table 3. Datasets used for Barents-2.5 km model evaluation. The listed references include links to the repositories where data and details on sampling and data processing can be found.**

| Compartment | Variable | Description | References |
|---|---|---|---|
| Sea ice | Ice concentration (dimensionless) | Regional high-resolution sea ice charts Svalbard region | Dinessen & Hackett (2016) https://thredds.met.no/thredds/catalog/myocean/siw-tac/siw-metno-svalbard/catalog.html |
| | | AMSR2 sea ice concentration product from University of Bremen | Spreen et al. (2008) https://seaice.uni-bremen.de/data/amsr2/asi_daygrid_swath/n6250/ |

**Table 4. Ice concentration values and their categorization used for the Ice charts and Barents-2.5 km model validation.**

| Ice concentration values | Re-mapped values |
|---|---|
| <0.01 | 0 |
| 0.01-0.1 | 0.05 |
| 0.1-0.4 | 0.25 |
| 0.4-0.7 | 0.55 |
| 0.7-0.9 | 0.80 |
| >0.9 | 0.95 |

### 2.3.2 S4K model

Datasets used for model evaluation are listed in Table 5, with links or citations to the various data sources. These include
ocean, sea ice and snow data. We used satellite products and *in situ* data collected during the N-ICE2015 expedition (Granskog
et al. 2018 and Figure 1). Therefore, more detailed comparisons between observations and model results are given for 2015.
We also compare TOPAZ4 reanalysis (https://doi.org/10.48670/moi-00007) with S4K model outputs regarding ocean and sea
ice variables listed below and in Table 5. Ocean data is used here to evaluate the "context" for the sea ice simulations. It
includes vertical profiles obtained with a CTD and with a microstructure profiler during the N-ICE2015 expedition (Table 5).
We used satellite data of sea ice concentrations, from regional high resolution sea ice charts for the Svalbard region (the same
mentioned above for the Barents-2.5 km model), and for sea ice and snow thickness, from the European radar altimeter
CryoSat-2, generated at Alfred Wegener Institute (AWI) for the winter period (October-April) (Hendricks & Ricker, 2020).
We also used Cryosat2-SMOS weekly Arctic sea ice thickness data (Ricker et al., 2017,
https://spaces.awi.de/display/CS2SMOS).
Sea ice plus snow thickness were collected during the N-ICE2015 expedition with a helicopter-borne electromagnetic
induction sounding (HEM) (King et al., 2016) and a ground based electromagnetic instrument (EM31) (Rösel et al., 2016a)
with footprints of approximately 50 m and 3-5 m, respectively (Haas et al., 2009). Snow thickness was measured with a
Magnaprobe with a footprint of approximately 0.2 m (Rösel, 2016b).

Table 5. Datasets used for S4K model evaluation. The listed references include links to the repositories where data and details on sampling and data processing can be found. CTD – conductivity-temperature-depth; MSS90L – Ocean microstructure profiler; HEM - helicopter-borne electromagnetic induction sounding; EM31 - ground based Electromagnetic instrument.

| Compartment | Variable | Description | References |
|---|---|---|---|
| Ocean | Practical salinity (psu) <br> *In situ* temperature (°C) | N-ICE2015 ship-based CTD and ocean microstructure profiles (MSS90L) | Dodd et al. (2016) and Meyer et al. (2016) for CTD and MSS90L data, respectively. |
| Sea ice | Ice concentration (dimensionless) | Regional high-resolution sea ice charts Svalbard region | Dinessen & Hackett (2016) |
| | Ice and snow thickness (m) | Arctic sea ice freeboard and thickness from the European radar altimeter CryoSat-2 | Hendricks & Ricker (2020) |
| | | Cryosat2-SMOS weekly Arctic sea ice thickness data | Ricker et al. (2017), |
| | | HEM, EM31 and Magnaprobe data collected during the N-ICE2015 expedition (Granskog et al., 2018) | King et al. (2016) for HEM, Rösel (2016a and b) for EM31 and Magnaprobe data, respectively. |

## 2.4 Model simulations

Simulations carried out with the Barents-2.5 km model are short-term, in accordance with its operational nature. Model evaluation was based on idealized simulations and on operational simulations and focused on sea ice concentration, which is the main variable of interest for this model. In the case of the S4K model, ~one-year simulations were carried out and comparisons between model and observations were focused on sea ice concentration, ice and snow thickness. Moreover, comparisons for the oceanic variables were also carried out.

## 2.4.1 Barents-2.5 km model

Model experiments with idealized wind forcing have been conducted with the Barents-2.5 km model in order to visually showcase the effects of using time-varying boundary conditions. The model was initialized from TOPAZ4 fields at 2019-09-01 and it ran until 2019-09-20. One run without the time-varying boundaries (just like the operational model ran before) and one with the boundaries extracted from TOPAZ4 results for the same period. All aspects of the model run, except the wind

forcing, were realistic. The wind forcing was idealized to be purely in the model xi-direction, positive in the first part of the run and negative in the latter part of the run. The goal was to blow the sea ice away from the left-most boundary before reversing the wind and observe the interaction with the boundary when the sea ice is forced towards it again. More specifically, the wind forcing was:

$$U_{wind} = \begin{cases} 10.0 \ ms^{-1}, t \leq 2019.09.07 \\ -10.0 \ ms^{-1}, t > 2019.09.07 \end{cases}$$

We also compare results obtained with operational simulations before and after the time-varying boundaries were introduced. These contrasting results are also evaluated against the satellite data. The operational model is initialized with data from TOPAZ4. We began using time-varying boundary conditions in the operational forecasts in October 2019 after spinning up the model for one month.

### 2.4.2 S4K model

The model was initialized from TOPAZ4 fields and ran from January 2014 until July 2015. Results were analyzed only from October 2014 after some spin-up time. Model output was compared with observations of ocean and sea ice variables measured in situ during the N-ICE2015 expedition (Granskog et al., 2018). Here we focus only on the evaluation of hydrographical properties with depth and on temperature-salinity diagrams. The satellite data was used mainly for evaluation of sea ice concentration and sea ice + snow thickness (Table 5). Comparisons were also made with TOPAZ4 results since it is an operational system in use by the Copernicus Marine Service (https://marine.copernicus.eu/) and it provides S4K sea ice boundary conditions. Ocean boundary conditions were from the Pan-Arctic A4 model described in Hattermann et al. (2016). The decision of using ocean boundary conditions from one model and sea ice boundary conditions from another one was based on results from preliminary simulations using only TOPAZ4 ocean and sea ice boundaries. The results of these simulations produced an unrealistically weak West Spitsbergen Current and large salinity and temperature ocean biases (not shown). Therefore, we tried using ocean boundaries from the A4 model which led to a significant improvement in our results.

### 3. Results

### 3.1 Barents-2.5 km model

### 3.1.1 Idealized simulations

The idealized simulations (results available at: https://zenodo.org/record/4727865#.YOMasRHis2w) show that when time-varying boundaries are not considered, and the wind direction is perpendicular to one of the boundaries a gap is created between

the ice edge of the Barents-2.5 km domain and the boundary with the TOPAZ4 domain (Fig. 2a and b). Moreover, when the wind is reversed, ice piles up at the boundary where the gap was formed, artificially increasing sea ice thickness. These "non-realistic" behaviors disappear once time-varying boundaries are considered, resulting in a relatively smooth transition between the results of TOPAZ4 and those of the Barents-2.5 km model (Fig. 2). This transition is not perfect, and signs of a "seam" can be seen where the external fields have been propagating through the boundary.

### 3.1.2 Operational simulations

Results from these simulations are available at: https://zenodo.org/record/4728069#.YOMLDhHis2w). The upper left panel of Fig. 3 shows typical modeled sea ice concentration fields prior to the usage of time-varying boundary conditions. While the overall field has a lot of details in each panel, there are significant artifacts, especially, along the top boundary. Northeastern winds force ice away from the boundary, leaving open water behind (Fig. 3a), creating an artificial polynya in the Barents-2.5 km. This was a regular occurrence in the original operational model. Fig. 3b shows the day before time-varying boundaries (OBC) were enabled. The more realistic ice field here resulted from reinitializing the model in 2019-09-03 with the TOPAZ4 fields after results shown in Fig. 3a and prior to results shown in Fig. 3b. This was done because the model had severely diverged from the observations. Fig. 3c shows the day the OBC fields were put into operation. This represents the one-month spun-up fields from TOPAZ4, while using time-varying boundary conditions, and immediately exhibits better correspondence with the external fields. Note that, at this point, this is a combined effect of the proximity (in time) to the re-initialization from TOPAZ4, and the effects of the new OBC's. That is why there is such a significant difference over only 1 day. It would have been a lot smaller had the OBC's been put into operation without a spin up run. Finally, Fig. 3d shows the situation after four months of running with the time-varying boundaries (before AMSR2 assimilation was put into operation). We observe a much better agreement between ice fields of TOPAZ4 and those of Barents-2.5 km models.

Figure 4a shows the Root Mean Square Error (RMSE) of the predicted sea-ice concentration from March 2019 to April 2021 in the operational Barents-2.5 km model calculated against AMRS2 and Svalbard ice chart observations, which tracked the performance of the operational Barents-2.5 km in the early two years. The vertical red line indicates the time when applying the time-varying boundaries, and the vertical green line shows the time when applying the data assimilation (see 2.3.1). Before the time-varying boundaries, the RMSE was generally between 0.2 and 0.4 (before mid-August 2019). Due to the large error in the open boundaries, the initial conditions had to be reinitialized in late August and September, which is seen in the abrupt decrease of the RMSE. However, the RMSE increased rapidly after each reinitialization. After implementing the time-varying boundaries in October 2019, the average RMSE is generally below 0.25, much lower than in the previous period. To further analyze the effect of the time-varying boundaries, we computed Taylor diagrams (IPCC, 2001; Taylor, 2001), using the MatLab PeterRochford-SkillMetricsToolbox-d7ea0d3. The improvement in model performance was negligible when the daily total sea-ice extent was considered (Fig. 4b). However, a large improvement is apparent when spatially resolved data are compared

(Fig. 4c), with higher correlation coefficient and lower RMSE for the simulation with time-varying boundaries. Moreover, the

model standard deviation becomes very close to that of the data. Altogether, this shows that the model accuracy improved, and

that ice concentration variability is better captured.

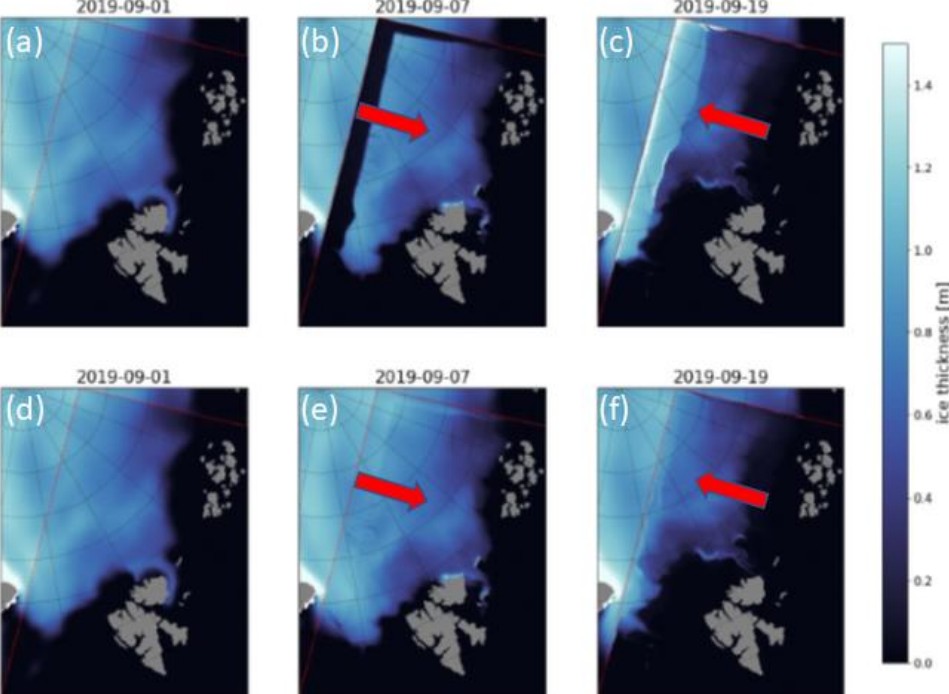

**Figure 2. Wind-idealized experiments with the Barents-2.5 km model plotted inside the TOPAZ4 model. The Barents-2.5 km model**
**was run in its full state except the wind forcing was idealized in the sense of constant wind in the model xi-direction. The figure**
**shows sea ice thickness fields at three moments in time for the run without (upper row) and with (lower row) time-varying**
**boundaries. The first column is the initial TOPAZ4 field interpolated onto the Barents-2.5 km grid, the second column corresponds**
**to Barents-2.5 km results after 6 days as the wind turns back in the negative direction, i.e. when the sea ice should be at its maximum**
**displacement relative to the left-most boundary, and the final column shows the state towards the very end of the run when the wind**
**has been blowing "left" for 12 days. Wind direction is shown by the red arrows.**

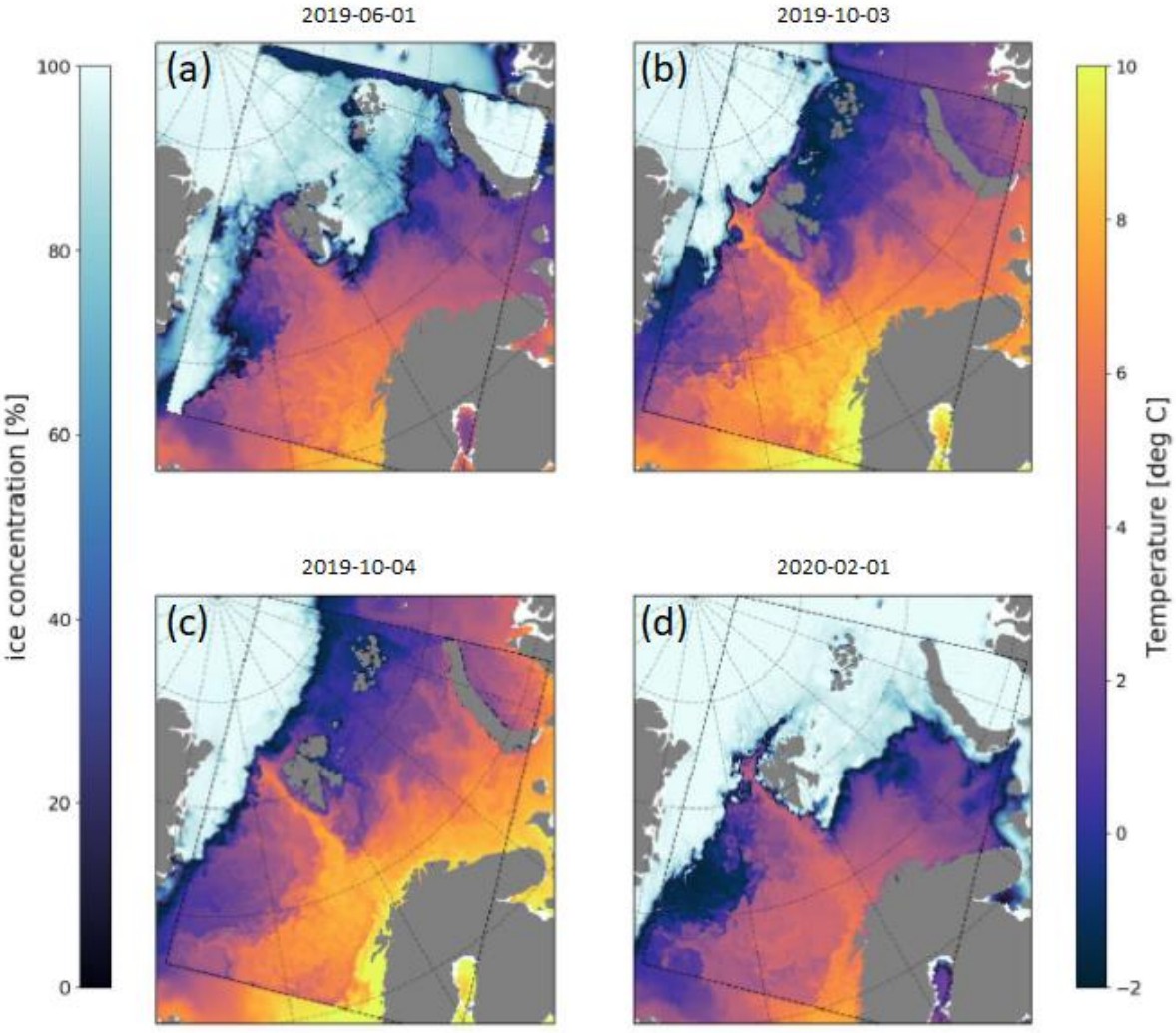

Figure 3. Operational simulations with the Barents-2.5 km model, plotted inside the TOPAZ4 model. These plots are taken directly from the operational model at MET and illustrate the effects of time-varying boundary conditions in the operational model. Sea ice concentration and surface water temperature fields (in the open water areas) are shown for three different dates at 00:00 UTC. Panel a) are a few months before new BC's, b) the day before new BC's, c) the day of new BC's and d) a few months after new BC's.

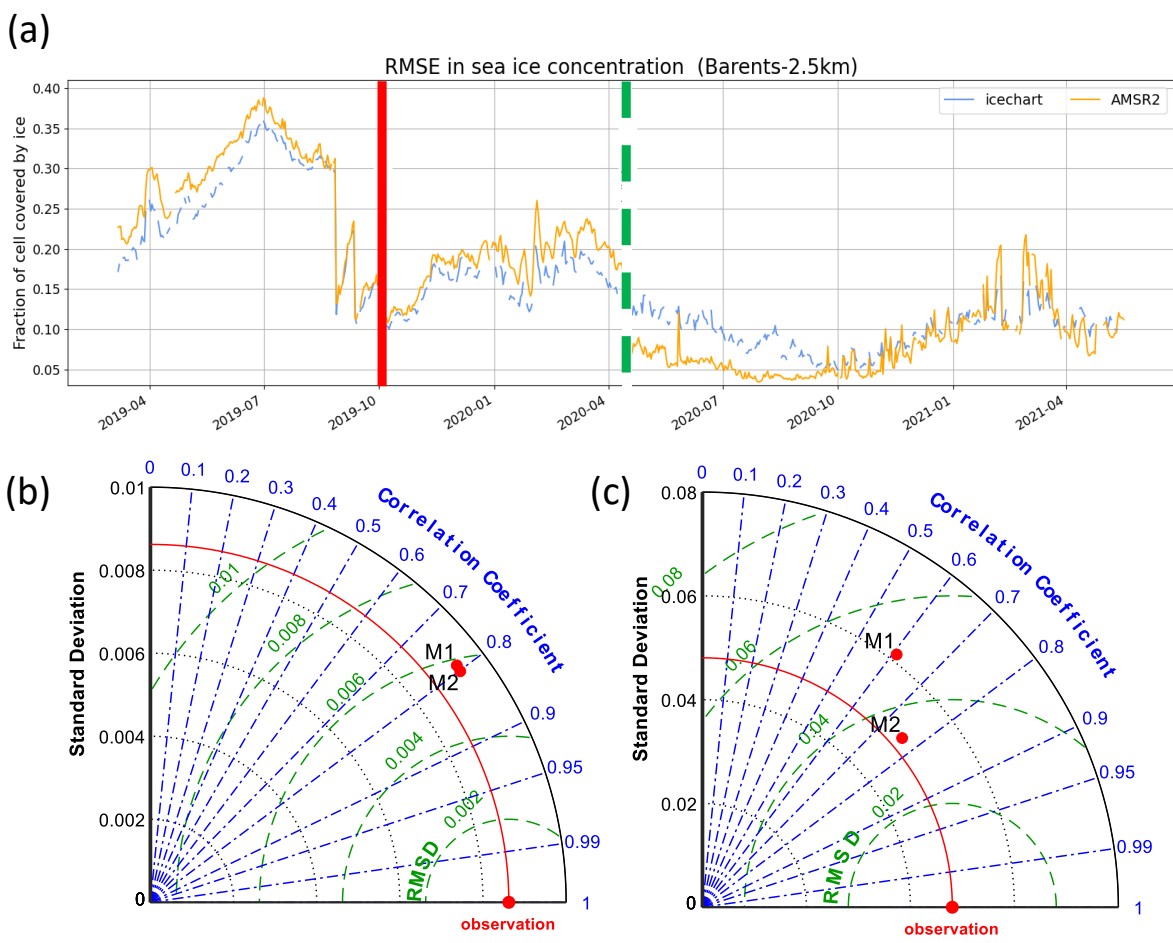

**Figure 4. (a) Root Mean Square Error (RMSE) of the Barents 2.5 km model for sea ice concentration, before and after using time-varying boundaries (vertical red line) and before and after data assimilation began (dashed vertical green line), calculated against AMRS2 and Svalbard ice chart observations (see 2.3.1). Lower panels: Taylor diagrams for the operational Barents-2.5 km simulations and AMRS2 observations, without (M1) and with (M2) the time-varying boundaries; (b) Daily results averaged over the whole model domain; (c) spatially resolved daily results. The red line in the Taylor charts depicts the standard deviation of the observations. The green curved isolines show the RMSE and the correlation coefficient is shown in blue.**

**3.2 S4K model**

We present first results for ocean variables and then for sea ice variables. In both cases we compare S4K with TOPAZ4 results and with observations (cf. – 2.3.2).

### 3.2.1 Ocean results

Extreme median salinity and temperature biases are ~-0.3 and -4 ºC and, ~+0.2 and -1.5 ºC, for TOPAZ4 and S4K, respectively (Figs. 5 and 6). The salinity biases within the top 100 m are smaller for TOPAZ and less than +0.2 ºC for S4K. The temperature biases within the same depth range are smaller for S4K. Both model bias for salinity and temperature are larger between c.a. 100 and 300 m than for the other depth ranges (Figs 5 and 6), being smaller for S4K than for TOPAZ. Temperature-salinity diagrams show better similarity between S4K and observations than between TOPAZ4 and observations (Fig. 7). Salinity and temperature ranges from S4K compare well with those of the observations (Fig. 7a *versus* Fig. 7b). In the case of TOPAZ, both ranges are much narrower than those of the observations (Fig. 7a *versus* Fig. 7b).

### 3.2.2 Sea ice results

Sea ice concentration and sea ice plus snow thickness from satellite products, TOPAZ4 and S4K show similar patterns (Figs. 8 and 9). In Fig. 8e and f and 9d, we plot S4K fields within a rectangle defined by a dashed line and "surrounded" by TOPAZ4 fields to evaluate the transition from TOPAZ4 forcing to the S4K fields. Boundary effects resulting from forcing S4K with TOPAZ4 sea ice data are not visible in the sea ice concentration plots (Fig. 8e and f) and they are quite smooth in the sea ice + snow thickness plots (Fig. 9d), with the exception of thinner ice along the North-East boundary in January 2015 (Fig. 9d). In some occasions, S4K predicts thin ice south eastwards of Greenland to a larger extent than observed in satellite data, and protruding from the ice flowing along Greenland and out of the Fram Strait (Figs. 8f and 9d). This is neither visible in the satellite data, nor in TOPAZ4 results (Figs. 8 and 9).

Sea ice + snow thickness results from S4K model are generally lower than those from satellite products and TOPAZ4 results for the overlapping areas (Fig. 9). However, sea ice + snow thickness frequency histograms based on EM31 data (Table 5) overlap more with S4K than to TOPAZ4 (Figure 10a and b). A similar comparison based on HEM data shows similar trends (Figure 10c and d). Despite the much smaller footprint of both the EM31 (3-5 m) and HEM data (50 m) (cf. – 2.3.2) compared with the model resolution (12.5 km for TOPAZ4 and 4 km for S4K), the observed sea ice + snow thickness ranges are much larger than those predicted by both models. Regarding snow thickness based on Magnaprobe data, both models have a negative bias (Figure 10e and f) and larger in absolute value than that for sea ice + snow thickness.

Here we show only a limited number of results due to space constraints. However, monthly averaged map plots of sea ice concentration and sea ice plus snow thickness, from the satellite products listed in Table 5, and from TOPAZ4 and S4K for the period August 2014 - July 2015) may be found at: https://doi.org/10.5281/zenodo.5800110.

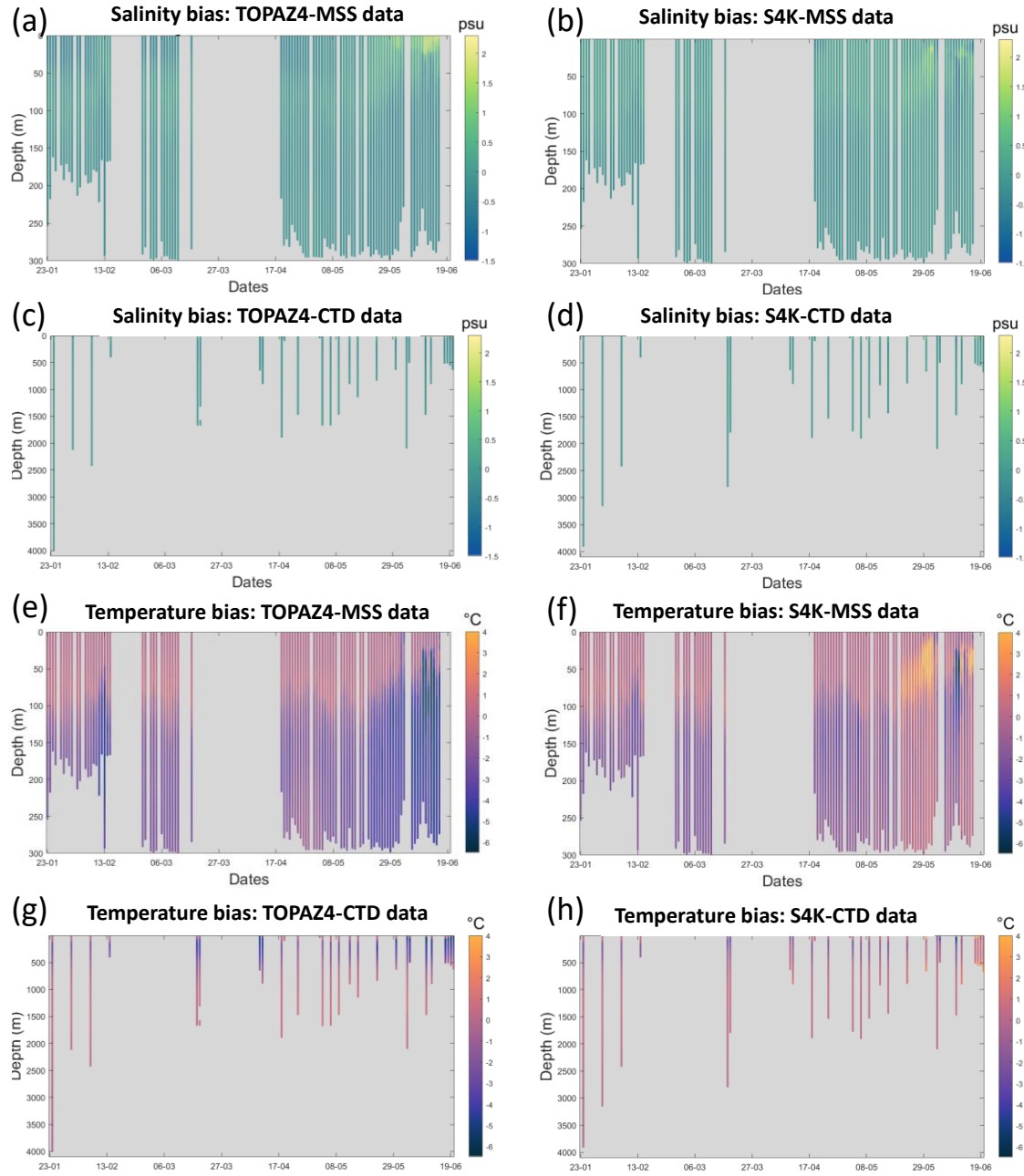

**Figure 5. TOPAZ4 [(a), (c), (e) and (g)] and S4K [(b), (d), (f) and (h)] model salinity (upper four panels) and temperature (lower four panels) biases, as a function of time and depth, from profiles obtained during the N-ICE2015 expedition (Granskog et al., 2018). Panels (a), (b), (e) and (f) show biases for the upper 300 m, based on data from ocean microstructure profiles (MSS) (Meyer et al., 2016). Panels (c), (d), (g) and (h) show biases for the whole water column, based on CTD profiles (Dodd et al., 2016) (see Fig. 1, Table 5 and text).**

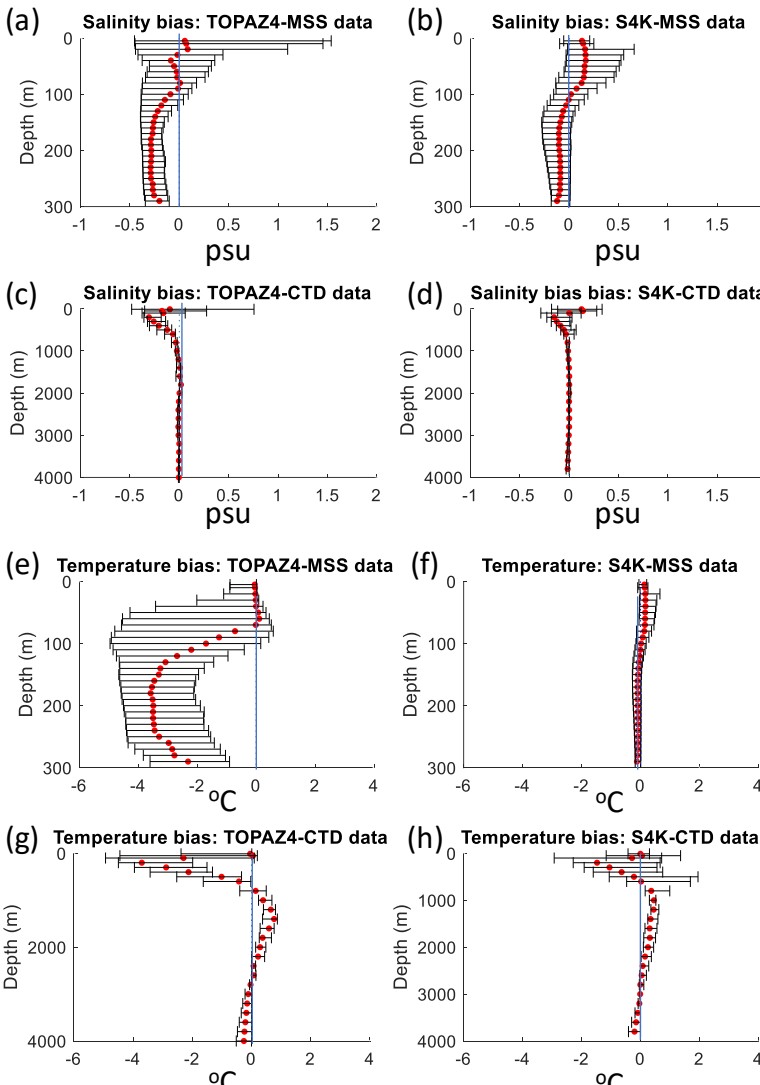

**Figure 6. Salinity and temperature median bias±10 and 90 percentiles for TOPAZ4 [(a), (c), (e) and (g)] and S4K [(b), (d), (f) and**
**(h)], as a function of depth, based on data obtained during the N-ICE2015 expedition (Granskog et al., 2018). Panels (a), (b), (e) and**
**(f) show biases for the upper 300 m, based on data from ocean microstructure profiles (MSS) (Meyer et al., 2016). Panels (c), (d), (g)**
**and (h) show biases for the whole water column, based on CTD profiles (Dodd et al., 2016) (see Fig. 1, Table 5 and text).**

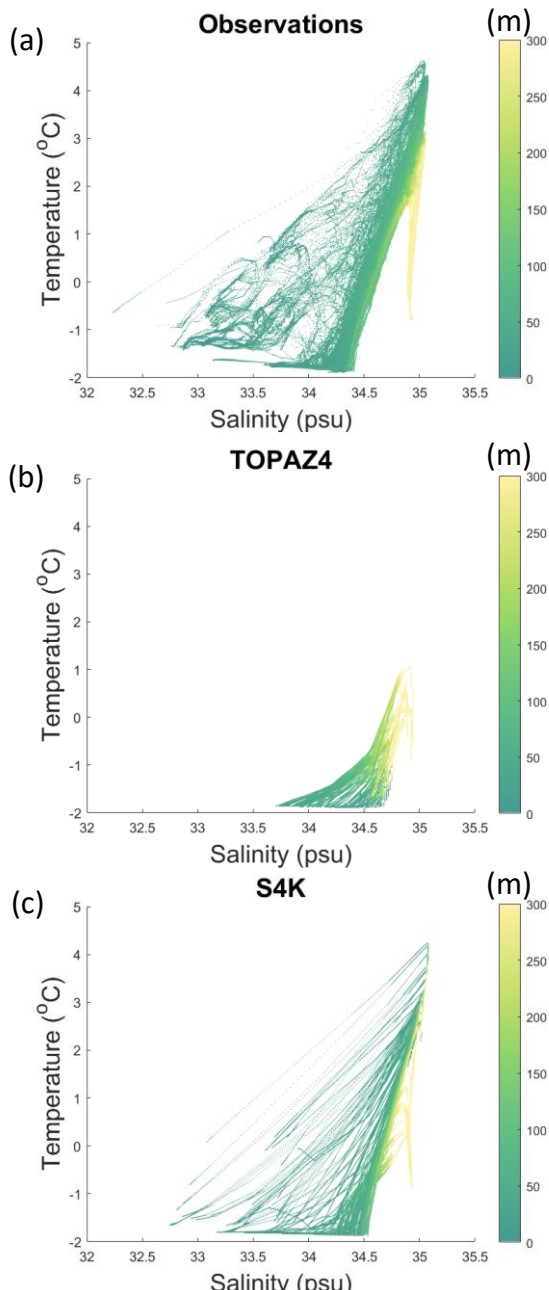

Figure 7. Temperature-salinity diagrams for observations collected during the N-ICE2015 expedition (Granskog et al., 2018) (a),
TOPAZ4 and S4K models for the same periods and locations as the observations [(b) and (c), respectively]. The color scale represents
depth in meters (see Fig. 1, Table 5 and text).

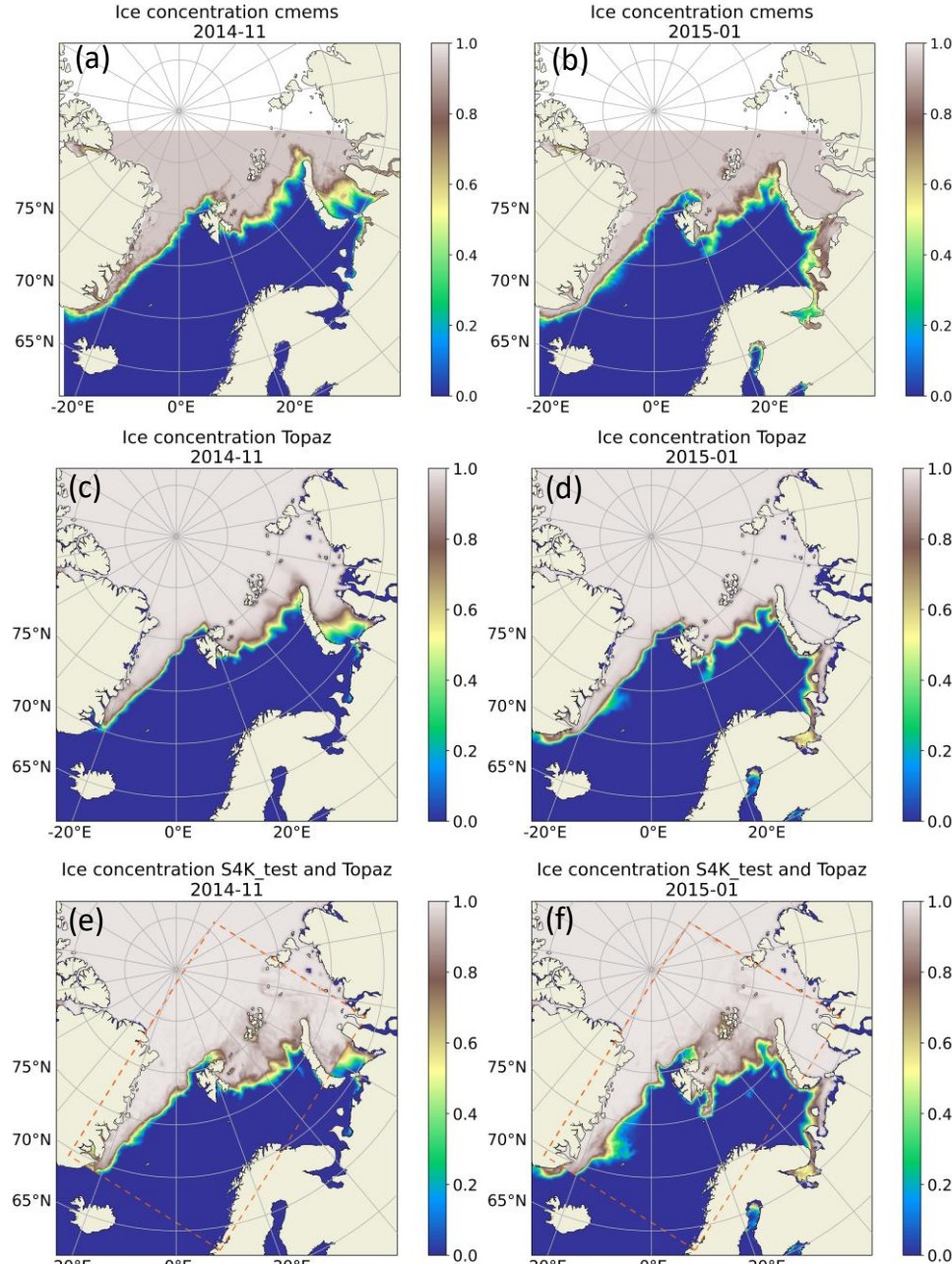

Figure 8. Dinessen & Hackett (2016) CMEMS (SEAICE_ARC_SEAICE_L4_NRT_OBSERVATIONS_011_002) [(a) and (b)],
TOPAZ [(c) and (d)] and S4K ((e) and (f)] results for monthly mean sea ice concentration fields for November 2014 (left panels) and
January 2015 (right panels). S4K fields are inserted in the TOPAZ4 model domain in the rectangle defined by the dashed line
included in panels (e) and (f) (see text).

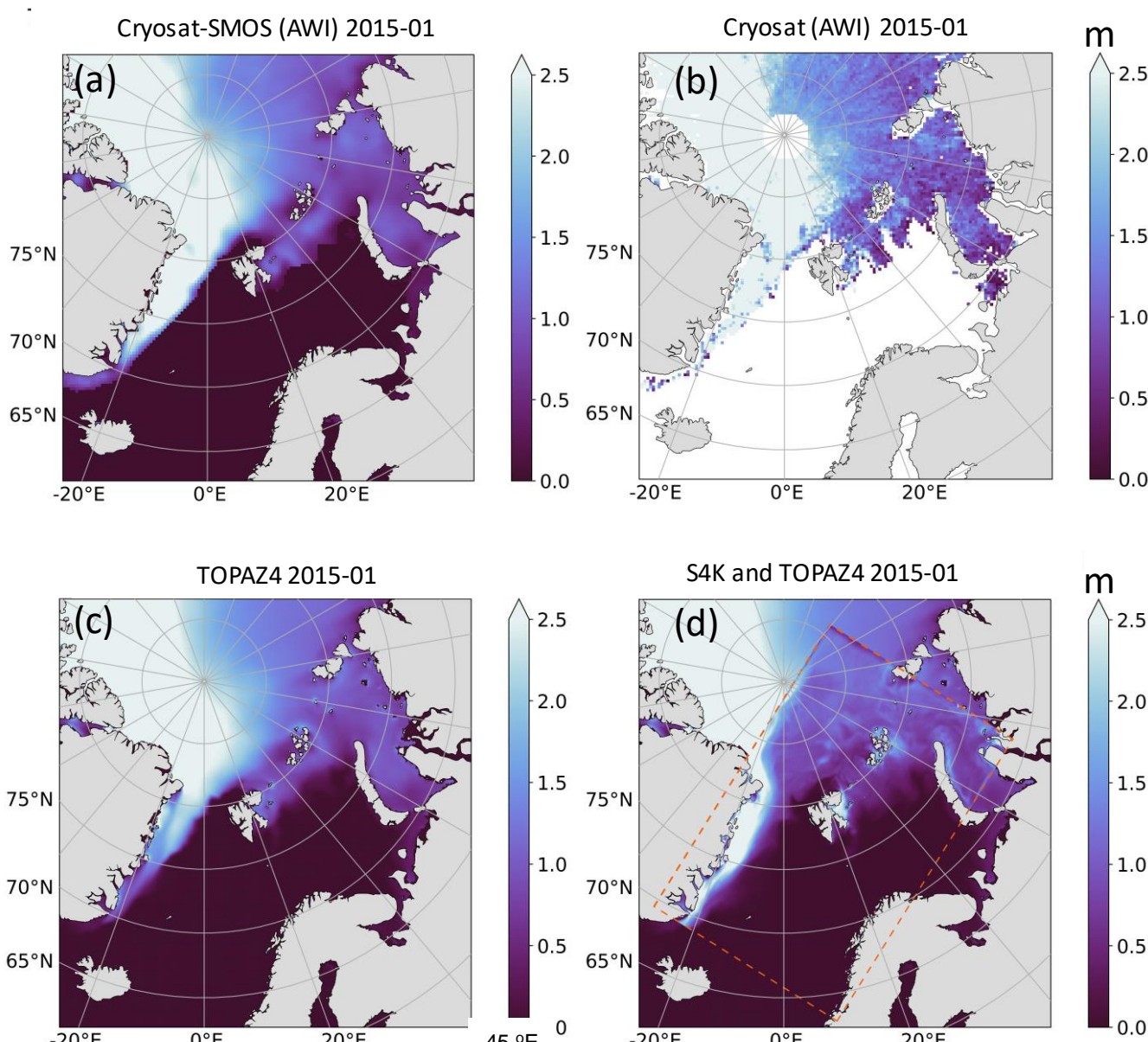

**Figure 9. Cryosat2-SMOS (a), Cryosat-2 (b), TOPAZ4 (c) and S4K (d) monthly mean sea ice + snow thickness for January 2015. S4K fields are inserted in the TOPAZ4 model domain in the rectangle defined by the dashed line included in panel (d) (see text).**

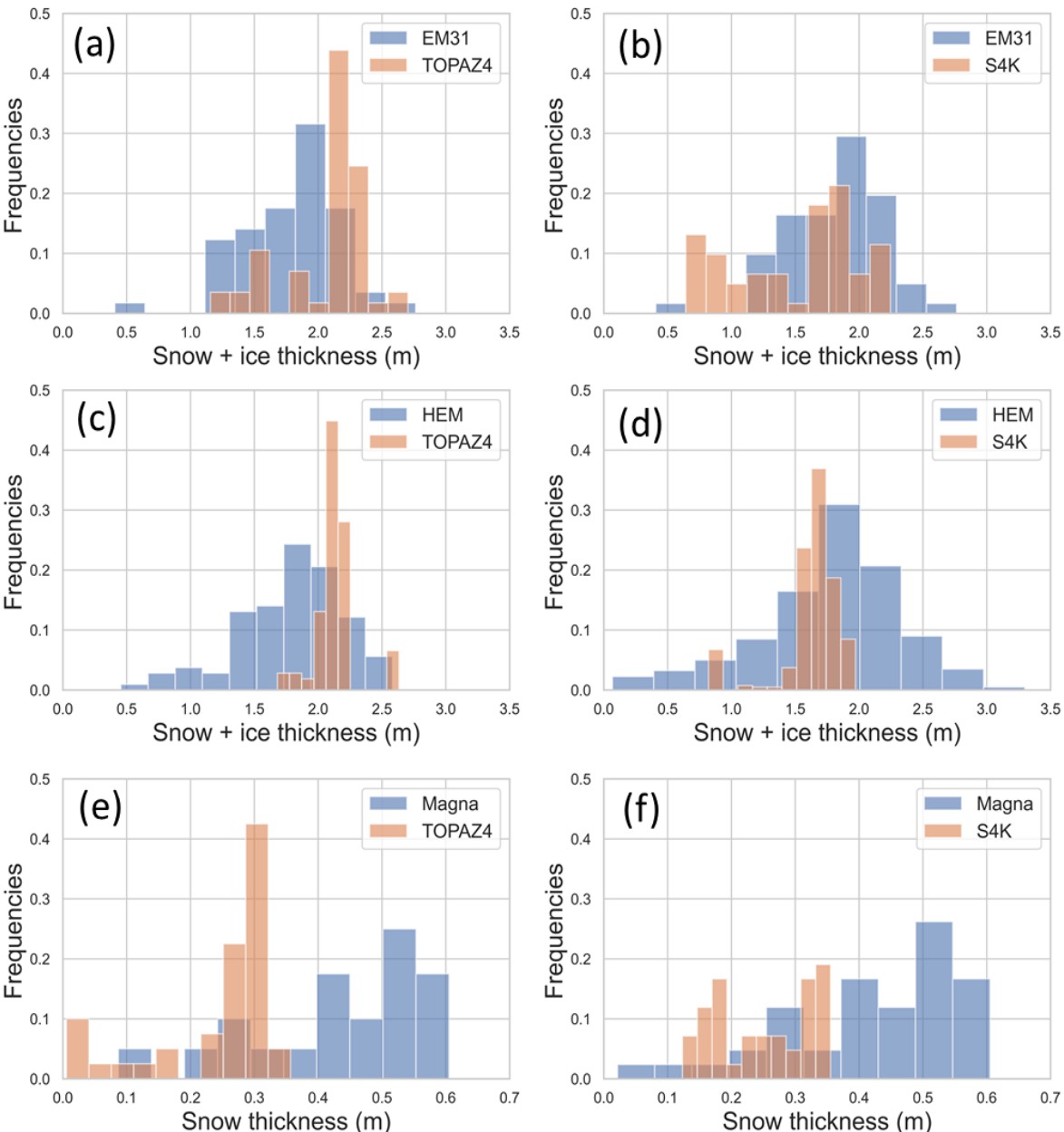

Figure 10. Observed (blue) and modeled (brown) frequency distributions of snow + ice thickness [(a)-(d)] and only snow thickness [(e) and (f)]. Measurements were taken during the N-ICE2015 expedition with the instruments indicated at the top of the panels: EM31 [(a) and (b)] and HEM[(c) and (d)], for snow + ice thickness and Magnaprobe [(e) and (f)] for snow thickness. Observational data were averaged for TOPAZ4 (left) or S4K models cells located in the same areas, resulting in slightly different observed frequency distributions, given the different spatial resolution of the models (12.5 and 4 km, respectively). Model results, averaged for the same areas and days where measurements took place, in the left panels are from TOPAZ4 and, in the right panels are from S4K (refer to Table 5 and text).

## 4. Discussion

The implementation of time-varying boundaries both in the Barents-2.5 km and the S4K models, resulted in a generally smooth transition between the fields of TOPAZ4, providing the boundary conditions, and the fields of the former two models. The performance of the operational Barents-2.5 km improved significantly with the usage of time-varying sea ice boundaries. This upgraded performance was also a large contributor to the Barents-2.5 km operational forecasts being more widely adopted in downstream applications like drift models and vessel icing models and as support for a specific ship salvage operation near Svalbard. There is a large demand for more realistic operational forecasts to support search and rescue, oil spill and other similar scenarios in the Barents Sea. The implementation of a more realistic boundary treatment for sea ice is a central step to achieve a wider usage of the operational fields.

Notwithstanding these results, we still can see some "seams" between the TOPAZ4 fields and those of the other two models. For example, some ice + snow thickness "artifacts" are visible in the S4K model results, especially in the Northeastern border of its domain (Fig. 10d). These "artifacts" may arise from drift differences inside the domain and at the boundaries. Such artifacts were already noted in the Barents-2.5 km model (refer to 3.1.1). Another problem is the different horizontal spatial resolutions of TOPAZ4 (12.5 km) and the models described herein (2.5 and 4 km). Perhaps the more likely explanation is the mismatch between available TOPAZ4 sea ice fields and those required by CICE (refer to 2.2.2 Model boundary data details). Recall from section 2.2.2 that extensive assumptions had to be made in order to fit the limited TOPAZ4 data for all the boundary variables required by CICE. In fact, experiments (not shown) done with a higher resolution model (500 m horizontal resolution) implemented with CICE, nested in the Barents-2.5 km model, and using exactly the same sea ice data of the larger model, did not show any seam but instead, a near perfect transition between both domains. This shows the importance of coordinating the storage of adequate outputs from larger models with the "needs" of regional models. The ideal output from a larger model should include the variables listed in Table 2 (corresponding to the variables defined to store boundary values), use the same sea ice thickness categories of the nested model and the same number of sea ice and snow layers.

In the tests carried out so far, we "relaxed" only the halo zone (more specifically, the grid cells surrounding the domain) and their neighbor cells to follow exactly the way CICE deals with boundary conditions. The default value in CICE for the thickness of this zone is one cell. In fact, this halo zone includes not only the domain boundaries but also the boundaries of all blocks of cells used in a parallel simulation. However, the boundary code affects only the cells surrounding the domain. A more complex treatment involving a broader relaxation zone with more than one cell thickness may be considered but it is out of the scope of the present study.

The S4K model has a smaller ocean temperature and salinity bias than that of TOPAZ4, in the region north of Svalbard, where the N-ICE2015 expedition took place (Granskog et al., 2018). Observed biases are larger at the depth range where Atlantic Water and Modified Atlantic Water are found (Meyer et al., 2017). There is a better fit between TOPAZ4 results and satellite data than those of S4K, which may partly result from the data assimilation process of the former. "Spurious" thin sea ice

predicted by S4K south eastwards of Greenland (cf. - 3.2.2 and Figs. 8f) results from the placement of the front between the

inflowing Atlantic Water and the Outflowing Polar Surface Water (e.g. Våge et al., 2018). In the S4K model, this front is not

close enough to east Greenland on some occasions, allowing very cold surface water to spread towards Svalbard, with

production of some thin sea ice.

As a final note we emphasize here the compatibility of the changes described in this study with the most recent versions of the

Los Alamos Sea Ice Model (CICE + ICEPACK, ), since the files changed and listed in Table 1 are similar to those of the most

recent versions.

## 5. Conclusion

We implemented time-varying sea ice boundaries in the Los Alamos Sea Ice Model (CICE). This implementation was tested

using two regional coupled ocean-sea ice models, both covering a large part of the Barents Sea and areas around Svalbard: the

Barents-2.5 km, an operational forecast model, and the S4K, a model used for research purposes. Sea ice boundary conditions

were obtained in both cases from TOPAZ4 - a well-tested and documented assimilative coupled ocean and sea ice model

covering the Arctic and North Atlantic oceans. Obtained results show significant improvements in the performance of the

Barents-2.5 km model after the implementation of the time-varying boundary conditions. The performance of the S4K model

in terms of sea ice and snow thickness is comparable to that of the TOPAZ4 system. The implementation of time-varying

boundary conditions described in this study is similar regardless of the CICE versions used in different models. The main

challenge remains the handling of data from larger models before its usage as boundary conditions for regional/local sea ice

models, since mismatches between available model products from the former and specific requirements of the latter are

expected, implying case-specific approaches and different assumptions. Ideally, model setups should be as similar as possible

to allow a smoother transition from larger to smaller domains.

## Code availability

The software code used in this study for the Barents-2.5 km model may be found at:

https://zenodo.org/record/5067164#.YOMK4hHis2w.

The ocean modeling code is a ROMS branch. Code licensing may be found at:

http://www.myroms.org/index.php?page=License_ROMS.

The software code used in this study for the SA4 model may be found at: https://doi.org/10.5281/zenodo.5815093

## Data availability

Results from the Barents 2.5 km model may be found at: https://zenodo.org/record/4727865#.YOMasRHis2w and https://zenodo.org/record/4728069#.YOMLDhHis2w, for the idealized and for the operational simulations, respectively, described in 2.4.1.

Graphical sea ice and snow results from the TOPAZ4 and S4K simulations may be found at: https://doi.org/10.5281/zenodo.5800110

## Authors contribution

Pedro Duarte made the first version of software changes related to the implementation of time-varying boundaries in the CICE code and ran the simulations with the S4K model.

Jostein Brændshøi, Yvonne Gusdal and Nicholas Szapiro implemented, tested and adapted those changes in the Barents-2.5 km model and ran the simulations shown in the paper with this operational model.

Dmitry Shcherbin performed software development and implemented and tuned the S4K model.

Pauline Barras processed and helped analyze S4K model results.

Jon Albretsen contributed to the analysis of the S4K model results.

Annette Samuelsen provided the boundary conditions from TOPAZ4.

Keguang Wang prepared the AMSR2 sea ice concentration and its standard deviation and performed the data assimilation.

Jens Boldingh Debernard led and performed the implementation of the CICE-ROMS coupling in METROMS and contributed to discussions of the OBC implementation in Barents-2.5 km model.

All authors contributed to the writing of the manuscript.

## Competing interests

The authors declare that they have no conflict of interest.

## Acknowledgements

This work has been supported by the Fram Centre Arctic Ocean flagship project "Mesoscale physical and biogeochemical modelling of the ocean and sea-ice in the Arctic Ocean" (project reference 66200), the Norwegian Metacenter for Computational Science application "NN9300K - Ecosystem modelling of the Arctic Ocean around 440 Svalbard", the Norwegian "Nansen Legacy" project (no. 276730), the European Union's Horizon 2020 research and innovation program under grant agreement No 869154 via project FACE-IT (The future of Arctic coastal ecosystems –Identifying transitions in

fjord systems and adjacent coastal areas), the "Arktis 2030"-programme and the project "Vær- og havvarsling for arktisk miljøberedskap" funded by the Norwegian Ministry of Climate and Environment, and from the European Union's Horizon 2020 research and innovation program under grant agreement No 101003826 via project CRiceS (Climate Relevant interactions and feedbacks: the key role of sea ice and Snow in the polar and global climate system). We acknowledge the usage of CryoSat-2 satellite products from the Alfred Wegener Institute (AWI), publicly available under a Creative Commons Attribution 4.0 International (CC BY 4.0) license and the usage of Cryosat2-SMOS satellite products. The production of the merged CryoSat2-SMOS sea ice thickness data was funded by the ESA project SMOS & CryoSat-2 Sea Ice Data Product Processing and Dissemination Service, and data from DATE to DATE were obtained from AWI. We thank the U. S. Department of Energy's (DOE) Earth System Modeling Program for allowing the use of this version of the CICE model, which includes updated biogeochemistry parameterizations within a "column package" developed as part of the Accelerated Climate Model for Energy (ACME) project. Tuning and model validation work performed under the DOE Regional and Global Climate Modeling Program also contributed to these results.

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
