# Peer review of "Implementation and evaluation of open boundary conditions for sea ice in a regional coupled ocean (ROMS) and sea ice (CICE) modelling system"

_Geoscientific Model Development, 2022_

## Author Comment (AC2)

To begin with, we wish to thank the critics from the **Anonymous Referee #1**, which we considered in revising the manuscript. This document is an update to the previous one, sent some weeks ago, before we received the comments from Referee #2.

For the sake of clarity, we reproduce the comments from the referee and then we present our answers. We indicate the lines of the paper where changes listed below took place, when relevant, using as reference the revised manuscript without the tracked changes.

All implemented changes may be tracked down in the attached files with the revised manuscript.

**Reviewer comments and our answers**

**RC1: 'Comment on gmd-2022-23', Anonymous Referee #1, 09 Mar 2022 reply Duarte et al.: Implementation and evaluation of open boundary conditions for sea ice in a regional coupled ocean (ROMS 3.7) and sea ice (CICE 5.1.2) modelling system**

The authors implement a new coupled configuration of the ROMS regional ocean model and CICE sea ice models for two regional modeling domains around the Barents Sea and Svalbard for operational (Barents-2.5km) and research (S4K) purposes. The authors present results showing that using time varying sea ice boundary conditions for the CICE sea ice model improve regional model results. It does appear that there are improvements to the model results with the time varying boundary conditions. However, the manuscript does not fully explain some of the model set up that would be necessary to replicate results, nor do they fully explain results and in some cases seem to make conclusions not supported by the figures. I think the authors need to expand their descriptions and discussions in the paper before the manuscript should be accepted.

**Answer:** We note that some of the general comments made above by the referee are detailed below, under "Major concerns". Therefore, we also detail below how we addressed the issues raised by the referee. Here we briefly state that we added more detailed explanations about the modeling set up and also about the description and discussion of our results and related figures.

**Major concerns**

- The authors do not provide some crucial details about the model assumptions:
  - Line 174: What slightly negative value did you use? -1.8C?
  - Answer: We used -0.00001°C. This is now specified in the text (line 205).
    Please note that this is assumed at the snow or ice-atmosphere interface when air temperatures are > 0.
  - Line 175: What sort of interpolation? How did this work if there was snow?
  - Answer: Linear interpolation. The same temperature trend was assumed for snow and ice. Therefore, when snow was present its height was considered at the thickness of each ice layer. These clarifications were added to the revised manuscript (line 206).
  - Line 260: Did you test the mixing of initial conditions to see the impact on the solution?

- Answer: No. The initial conditions came all from TOPAZ4 and we let the S4K model spin up before comparing the results with observations collected in 2015.
- Line 401: I disagree with the statement that the lower resolution model needs the doesn't necessarily need the same number of categories and layers, but you do need information to interpolate from the original to the new model.
- **Answer:** Please note that this sentence refers to what we understand as "ideal conditions". Actually, the sentence starts with "The ideal output…". So, we are not talking about the necessary conditions where interpolations are certainly possible, as we did in this study, where boundaries were far from what we consider "ideal".
- Impact of time varying Boundary Conditions:
  - Line 287/Figure 4: You state that the time varying boundary conditions lead to lower RMSE. However, in Fig. 4a it looks like the drop in RMSE happens BEFORE the change in boundary conditions. What is going on here? It is also impossible in Fig 4b to see the difference in M1 and M2. Can you use different colors or shapes for the markers?
  - Answer: Yes, the drop in RMSE indeed occurs before the change in boundary conditions. That is due to the re-initialization from TOPAZ (based on 1-month spin up) as described in line 283 of the "old" manuscript version. Moreover, please note that the average RMSE is much lower in the period between the beginning of usage of time-varying boundaries (red vertical line in Figure 4a) and the beginning of data assimilation (green vertical line in Figure 4a) than in the previous period, despite the drop in RMSE mentioned above and related to model re-initialization. We added a few comments to the last paragraph of 3.1.2 to clarify these issues (lines 328-351). We also changed Fig. 4b and now the two red circles are visible despite their great proximity.
- Line 393: Is it possible that some errors are caused by the discontinuity at the boundary? Many models - e.g. WRF for the atmosphere (https://doi.org/10.1175/1520-0477(1997)078%3C2599:ATOLBC%3E2.0.CO%3B2) or RASM in the ocean (https://doi.org/10.3189/2015AoG69A760) – have some sort of buffer zone to help with spurious boundary issues. Did you investigate this at all? Is it worth investigating?
- **Answer:** Thanks for pointing this out. We have recently investigated this problem. It is true that such spurious boundary issues are mainly due to the imperfect ocean open boundary conditions, particularly when tide is activated (as the TOPAZ4 does not include tide). Such imperfect ocean boundary conditions will result in artificial boundary currents, transfer erroneous energy along the open boundary, thus affecting the sea ice there. We have tested some cases by changing the sponge zone settings, which mitigate the effect of this error. This work is underway and will be reported later in a following study. For the sea ice boundary conditions applied here, our experience with a higher resolution model (500 m horizontal resolution) implemented with CICE, nested in the Barents-2.5km model, and using exactly the same sea ice data of the larger model, showed a near perfect transition between both domains. This suggests

that model will perform robustly when the ocean boundary conditions are correctly treated, and the sea ice boundary conditions match the exact needs of the model. This was already referred in the manuscript following line 393 (in the old version) and we kept it in the revised version (lines 453-459).

- Line 320-322: Your description about the biases is missing a lot of the detail from Figure 5 and in some cases contradicts it. For example in Figure 5 shows that S4k biases are largest from the surface to 100m whereas your description states the opposite. You also need to explain how the agreement shown in Figure 7 is better – larger range, warmer temperatures, fresher waters?
- **Answer:** The temperature biases between the surface and 100 m is much smaller for S4K than for TOPAZ. The overall salinity biases within this depth range is smaller for TOPAZ but at the surface is smaller for S4K (Figures 5 and 6). The agreement in Figure 7 is better for S4K because both the salinity and temperature ranges compare well with those of the observations (upper panel versus lower panel). In the case of TOPAZ both ranges are much narrower than those of the observations (upper panel versus middle panel). We added text to 3.2.1 to detail better these results (lines 378-384).
- Figures 8, 9, 10: Can you zoom in on the relevant domain rather than show pan-Arctic figures. It's hard to see anything meaningful in the full Arctic scale when the important details are at the transition/boundary of the regional model. Also, making difference figures might help as well to highlight the differences you describe because at present the differences are challenging to see. Adding arrows or other markers may help too.
- **Answer:** Figures were zoomed as suggested.
- Figure 9: the boundary looks smoother, but why is there so much less ice in S4K (9d) than Topaz (9c)? This seems important and worth explaining. It seems the S4K model is performing worse with so little ice.
- **Answer:** Please note that Figure 9 corresponds to 2014 results when the model is spinning-up. We removed this figure following a suggestion from the other referee. However, comparisons with measurements done in 2015 in Figures 11 show a better correspondence between S4K and observations as pointed out in the paper.
- Figure 11: c/d. Why do the frequencies for HEM/EM31/Magna vary between the columns? Are they on the same domain, because if so they should be identical frequencies. Is it possible to put all three PDFs on one figure for the same domain?
- **Answer:** This is a result of the different resolution of TOPAZ and S4K. Observational data were averaged for TOPAZ4 or S4K model cells located in the same areas, resulting in slightly different observed frequency distributions, given the different spatial resolution of the models (12.5 and 4 km, respectively). This explanation was added to the revised manuscript in Figure 10 caption.

**Minor concerns**

- Figures 2 and 3: Please provide more descriptive, standalone captions.
- Answer: Done as suggested.

- Introduction: You should probably address some of the points from this paper on whether sea ice models from climate models should be used for forecasting (https://doi.org/1007/s40641-020-00162-y). Also at line 386-388 you mention relevant operational reasons this model improvement has been useful. It seems this should be expanded on more in the introduction.
- **Answer:** We added a paragraph to the Introduction (lines 48-54) citing the paper mentioned by the referee and addressing some of its points, as suggested. We also expanded a bit about the importance of the improvement in the operational forecasts in the first paragraph of the Discussion (lines 438-445).
- Line 35: You should provide a DOI for whatever version of CICE you are using. See the table here: https://github.com/CICE-Consortium/CICE/wiki/CICE-Release-Table..
- Answer: This release table does not seem to include CICE 5.1.2, but we cite the manual of CICE5.1 (Hunke, E. C., Lipscomb, W. H., Turner, A. K., Jeffery, N., Elliot, S.: CICE: the Los Alamos Sea Ice Model. Documentation 474 and User's Manual Version 5.1. Los Alamos National Laboratory, USA. LA-CC-06-012, 2015.). Moreover, we include the DOIs to the exact versions we used in the paper, that include our own modifications. In fact, we appreciate this comment very much because we realized we made a mistake here by forgetting to specify that the Barents 2.5 km model is indeed based on CICE5.1.2 but the S4K model is based on a "columnar" version of CICE that the first author received from the CICE consortium in 2017. Thus, it is not possible to identify it in the table cited above. Because of this, we removed 5.1.2 from the title since more than one version of CICE is used in this study. The alternative is to specify both versions in the title but since the columnar does not correspond to any of those listed in the link provided by the referee, we can refer to it only as "columnar" which is a bit vague. Most likely it was a developing version between releases. However, this makes no big difference with regard to the implementation of time-varying boundary conditions since, as explained in the last paragraph of the Discussion, the CICE files changed are common across different CICE versions including CICE + Icepack. We added some comments to 2.1.1 (lines 81-82) and 2.1.2 (lines 105-107) to clarify that different CICE versions were used in the Barents 2.5km and the S4K models. We also added some text to the Acknowledgements (lines 518-522) regarding the CICE columnar version used for the latter model and following the agreement with the CICE consortium. By the way, we also checked (https://github.com/CICE-Consortium/CICEsvn-trunk/wiki/CICE-Versions-Index-(older)) for CICE versions released prior to CICE6 but we did not find an exact match to the versions used herein.
- Line 37: You may want to clarify that CICE6 and later have Icepack separate from the dynamics but CICE5 and any earlier versions they were combined. Use the specific version numbers here.
- **Answer:** Done as suggested. Please see also our answer to your previous comment and the text right after 2.1 Model description (lines 65-75).
- Line 40: You should provide more information about other modeling systems that use CICE. even older beyond the CMIP6. RASM versions. E.g. (https://doi.org/10.3189/2015AoG69A760), CESM (https://doi.org/1029/2019MS001916), Canadian Center Operational Meteorological (https://doi.org/10.5194/gmd-2020-255), Danish Institute (https://doi.org/10.1002/2017JC013481), etc.

- **Answer:** Done as suggested. More citations and corresponding references added to provide examples of modeling systems that use CICE (lines 72-75).
- Line 68: It would be helpful to have 1 sentence about TOPAZ4 and why it's optimal for this set up since not all readers will be familiar with it.
- Answer: Done as suggested. A sentence added about TOPAZ (lines 88-90).
- Line 71: AMSR2 link doesn't work. Also maybe mentioning here that in section 2.4.1 you will describe how the satellite product at 6km is downscaled.
- **Answer:** We have just tested the link and it worked. We wonder what the problem might be. We added a small sentence following the suggestion made by the referee about downscaling details.
- Line 87: Is this run continuously or with restarts (as reanalyses are usually run)?
- **Answer:** Please note that this is not a reanalysis product. It is run without restarts unless something wrong happens, in which case restart files are used for a "perfect" restart. For the sake of clarity, we added "run continuously" (line 113) to the sentence. Please refer also the Introduction where we characterize both modeling systems and explain that while Barents 2.5km is an operational system, the S4K model is a research tool (lines 44-45).
- Line 133-135: Do you have an estimate about the extra computational cost for using more memory? 10% or 90% increase?
- **Answer:** We do not have such estimate, but it should be marginal (< 10%?) as compared to other files (especially ROMS boundary files, which are huge due to the many time slots) because the ice forcing routines read only two time slices of the boundary conditions for proper time interpolation. Therefore, we are talking about a few arrays (as many as the boundary variables) with dimensions equal to those of the grid X 2 time slots and, for some variables, X the number of ice layers.
- Line 161: there are two periods at the end of the sentence.
- Answer: Corrected.
- Line 166: use "internal" instead of "inner".
- Answer: Done as suggested.
- Line 294: Remove "likely" as you have not yet shown this to be the case. The sentences below do show this, but at this point it seems unproven.
- **Answer:** There is no "likely" in line 294.
- Line 406: There is no figure 10e.
- **Answer:** We deleted this wrong reference to a figure.

To begin with, we wish to thank the critics from the **Anonymous Referee #2**, which we considered in revising the manuscript. For the sake of clarity, we reproduce the comments from the referee and then we present our answers. We indicate the lines of the paper where changes listed below took place, when relevant, using as reference the revised manuscript without the tracked changes.

All implemented changes may be tracked down in the attached files with the revised manuscript.

**General comment from referee #2**

Comments to this manuscript is based on the original manuscript, thus there may be some changes according to the review by the first reviewer that I have missed. I have tried to avoid repeating the same points. The manuscript describes a new nested modeling system for ROMS and CICE. Resolution is important and computational expensive, thus nesting of high resolution areas into large scale models with higher resolution is of interest as well as a nesting capability within CICE. That being said I agree with reviewer 1 that some work still needs to be done in order to accept this manuscript both on the structure of the paper and on the My comments follows below:

The "ideal" boundary conditions are mentioned. Correct these are not available from remotely sensed data. Most likely this is not the case for any model output from a "none in-house" model system either. For instance Topaz4 only deliver ice concentration/ice thickness (and maybe a few more parameters) if downloaded from Copernicus. Some estimates have to be made here which also applies when assimilating data. This discussion and maybe also the result of "bad choices" should be at least mentioned and maybe discussed. In addition what would be needed in order to expand this to be a "cheap" assimilation scheme

**Answer:** We discuss this issue in the second paragraph of the Discussion (lines 446-449), focusing on matching problems between the boundary data and the requirements of the CICE model. The result of "bad choices" is certainly important but it is difficult to develop much on this because we would have to go through a trial and error study to look for them. However, we gained some experience about bad choices during the implementation of the boundary conditions. For example, initially, we did not force the boundaries with the velocity fields, assuming instead the velocities calculated by the model near the boundaries. We noted that this could cause important bias in sea ice drift inside the model domain. There was a great improvement by including boundary sea ice velocities from the larger model.

**Some of the conclusion would have been easier to diagnose if the model systems were run at the same time and not in different time periods.**

**Answer:** The results of both modeling systems are independent and analyzed on their own. Given their different purposes and contexts it was not practical to analyze them for the same periods.

Mayor

I would focus the abstract more on the results and challenges instead of listing all boundary condition parameters used and which models are used for boundary conditions.

**Answer:** Done as suggested. A sentence was added to the end of the Abstract to emphasize the main challenges of our approach, in line with the second paragraph of the Discussion. The list of variables included in the boundary conditions was removed.

Line 34 - 41 I think that this belongs to a model description, which makes the introduction very short. I would like to see a section of other attempts to use ice models in a nested configuration. Both CICE and other models.

**Answer:** We transferred the mentioned lines to 2.1 Model description, with some small changes in the text. We also added more text to the Introduction about sea ice models in a nested configuration (lines 55-62). However, we were not able to find any description about the implementation of time-varying sea ice boundaries for the CICE model. We found such implementation for The Louvain-La-Neuve sea ice model LIM3.6 and this is now referred to in the Introduction (lines 60-62). At the end of section 2.3.1 we point out the differences between our approach and that of LIM3.6 (lines 188-192).

More complex boundary conditions are mentioned. Have you considered the effect of relaxing towards a value in the halo zone only compared to relaxing in a larger area or a flux representation? This could be brought up in the discussion just as the "perfect" availability of data is.

**Answer:** We did not try relaxing to a larger area. We followed a similar approach as implemented in CICE when setting the boundaries to some predefined value and affecting only the "ghost cells" in the halo zone, which are the cells surrounding the domain, and their neighbor cells. The default value in CICE for the thickness of this halo zone is one cell. In fact, this hallo zone includes not only the domain boundaries but also the boundaries of all blocks of cells used in a parallel simulation. However, the boundary code affects only those cells around the domain, for obvious reasons. Regarding "flux representation", please note that all variables used along the boundaries are tracers, except for the sea ice velocities which are fluxes. Following the suggestion from the referee we included a few sentences in the Discussion (lines 460-465) about these topics to complement what was written in the 2.3.1 (lines 166-172).

The location of the N-ICE cruise would be nice. Maybe in figure 1.

**Answer:** Please note that the location of the cruise is already included in Figure 1 – the insert on the lower right corner. In the captions we write: "The insert at the right bottom corner represents Svalbard and the area where the various drifts (lines showing the begin and end dates of each drift) of the N-ICE2015 expedition (Granskog et al., 2018) took place and along which sea ice and ocean data detailed in Table 3 were collected.".

I would maybe skip one of the figures 9 or 10 as the information is somewhat redundant.

Answer: Done as suggested, Fig. 9 was removed.

S4K vs TOPAZ ice thickness. Could you elaborate a bit on why you think that S4k has a better ice cover (atm forcing, higher resolution ocean or?)

**Answer:** Please note that we do not think that S4K has a better ice cover. We actually see that TOPAZ4 ice concentration results are more similar to satellite data which is an expected outcome since TOPAZ4 assimilated ice concentration data. It is difficult to elaborate on this given so many possible reasons for the differences. We merely write that snow + ice thickness in situ data compares better with S4K than with TOPAZ4 but we have to remember that these data are only for the N-ICE2015 surveyed region. So, we cannot be sure about the relative performance of these two models which are fundamentally different with TOPAZ4 being an operational system and S4K being a regional model without data assimilation.

Minor

Line 19: "sea ice size categories". I would change to "sea ice thickness categories"

Answer: Done as suggested.

Line 27 Norwegian Meteorological Institute shortname is MET here and Met Norway a few lines down. This should be consistent.

Answer: Corrected.

Line 50 please reformulate "We have chosen to use these two models because, whereas....

Answer: "whereas" was removed.

Line 62 "(for details on coupling refer 2.1.3)" add "to" after refer

Answer: Done as suggested.

Line 63 vertical coordinate following the bathymetry = terrain following?

**Answer:** Yes. We replaced "vertical coordinates following the bathymetry" with "terrain-following vertical coordinates".

Line 69 TPXO reference, which version and reference? The same is the case for nve data (mentioned further down) and IBCAO. Reference should be mentioned first time they are used. There are likely more that.

**Answer:** Versions numbers added as well as associated citations and references. Please refer to lines 90-94.

Line 79 S4K is 4km in resolution. What is the resolution of the A4 model. 4 as well? If this is the case what is the motivation for nesting? Is A4 only ocean or what is the reason for not using this for both ocean and sea ice conditions

**Answer:** The resolution of the A4 model is also 4 km. The motivation for nesting was to evaluate the sea ice boundary implementation. Moreover, we use this smaller domain for coupled physical-biogeochemical simulations. Initially, we planned to use only TOPAZ4 results for both the ocean and the sea ice boundary conditions, motivated by its operational nature. However, we found out that when doing so, we got an unrealistically weak West Spitsbergen Current and large salinity and temperature ocean biases (please refer to 2.5.2, lines 309-312). Therefore, we have chosen to use the A4 model results for ocean boundaries. We also found out that the A4 model underestimated snow cover which prevented its usage for sea ice boundaries, whereas TOPAZ4 sea ice results are "corrected" with data assimilation and, therefore, expectably reliable when it comes to sea ice concentration. Therefore, we used TOPAZ4 results for sea ice boundaries.

Line 87. Why use ocean boundary conditions from A4 and ice from TOPAZ. This seems to be a place where inconsistencies can be found.

**Answer:** Please see our previous answer. This "inconsistency" resulted from the limitations described above and is referred to in 2.5.2. However, we do not think that it is a major issue in this context where we are focused on see how the sea ice model performs when using time-varying sea ice boundary conditions. Therefore, we need the best possible ocean simulation.

Line 184 depth of layer ... Should this have been the fractional depth of each category?

**Answer:** This is the fractional depth of each layer and it is indeed calculated for the different categories. However, since the boundary data originating from TOPAZ is not discretized into ice thickness categories, this was done for only one category following what we wrote in point 2) under heading 2.3.2 (lines 201-203).

Line 192 Microwave -> Passive Microwave

Answer: Corrected.

Line 193: The ice charts are drawn manually as polygons. The resolution is the gridding resolution not the actual resolution.

**Answer:** "higher spatial resolution" was replaced with "gridding resolution" (line 225).

Line 320 ocean results

Is the improved results due to the A4 ocean boundaries, the ice boundaries or? It would have been easier to distinguish if Topaz4 were used all the way through. Line 320 states that the bias is larger for depths between 100 and 300m. It does not say compared to what. I assume that this is other depths.

**Answer:** Yes, the improved ocean results are due to A4 ocean boundaries, when considering the results obtained with TOPAZ4 boundaries (not shown in the paper but referred to). We added "than for the other depth ranges" (line 381).

On a general note I would say that three to four lines of text is very short for 3 big figures. I would evaluate some more on this result or leave it out and maybe add it to supplementary work.

**Answer:** Now we have some more lines. Moreover, these results are important to show that the ocean part of the model is doing relatively well and, therefore, is likely not the cause of major sea ice bias.

Line 330 (and 407) Any suggestion why the ice has a larger extent in S4K? I think that the assimilation is probably the cause as mention, however the S4K should have a resolution that improves the front. Maybe fresh water

**Answer:** As far as we managed to see, this small are of sea ice formation south eastward of Greenland is more due to thermodynamic sea ice formation than to sea ice advection. However, it is very thin ice and it seems to result from a slight sea surface temperature bias that may be due to problems with ocean mixing or with atmospheric forcing. We comment on this in lines 469-473

Table 2: I would put this in an appendix.

**Answer:** This table shows important technicalities regarding the implementation of sea ice time-varying boundaries and this is a technical development paper so we would prefer to keep it in the main text.

Figure 3 b and c. I am not entirely sure based on the text. Was figure 3.b just after starting from Topaz4 or did it run for a while and then it is just "luck" that they start so close to each other in figure 3b. Furthermore line 305 (figure 3 caption should be updated to four instead of three).

**Answer:** This was now clarified in the text. Please refer to the first two paragraphs of 3.1.2 Operational simulations.

Line 394 refer 2.3.2 . Is this in this manuscript?

**Answer:** Yes, it is a sub-section of Methodology.

Figure 4 (a)

I agree with the comments from reviewer 1 here. Another thing that puzzles me is why does the RMSE for the AMSR2 drop at the green line and start to increase again in winter time?

**Answer**: The drop in RMSE occurs before the change in boundary conditions. That is due to the re-initialization from TOPAZ (based on 1-month spin up) as described in line 283 of the "old" manuscript version. Moreover, please note that the average RMSE is much lower in the period between the beginning of usage of time-varying boundaries (red vertical line in Figure 4a) and the beginning of data assimilation (green vertical line in Figure 4a) than in the previous period, despite the drop in RMSE mentioned above and related to model re-initialization. We added a few comments to the last paragraph of 3.1.2 to clarify these issues.

We also changed Fig. 4b and now the two red circles are visible despite their great proximity. The drop of RMSE right after the green line was due to the beginning of data assimilation. We suspect that the increase in RMSE towards winter time it is related to Barents-2.5 km known cold sea surface temperature bias manifesting itself more in the winter.

Figure 11. Header of figure a,b and c,d seems to be swapped unless it is the caption text that is swapped.

Answer: Corrected.

---

## Referee Report (RR1)

Review of Implementation and evaluation of open boundary conditions for sea ice in a regional coupled ocean (ROMS) and sea ice (CICE) modelling system

This is the second review of the manuscript that demonstrates the value of boundary conditions that includes dynamics. The manuscript has improved however I still request some minor changes. One is the missing conclusion.

Abstract line 29. I assume that the improvement is due to the use of A4 oceanic boundary conditions rather than the nested area. If this is the case it should be mentioned here.

Line 51 and 52

"Also, knowledge about the possibility of ice in an area might be more important for applications than the specific details of the sea ice cover."

I am sure what is referred to here. Is it that the results are used as statistics of a hindcast for planning or is it that an approximate sea ice cover is good enough for some application? Please clarify

Line 68: CICE do include two packages from v6 and onwards, however you models are version 5.1.2 and something close to v6.0.0 (I assume). I am not sure whether it add value to mention it or if it confuses more that you mention it.

Line 77 primary model for forecasting **of** sea ice conditions….

Line 82 should this refer to section 2.2?

Line 86 could you add a reference to Arome-Arctic

Line 105 Here you could referebce the cice 6.0.0alpha CICE Release Table · CICE-Consortium/CICE Wiki · GitHub as the code you got is likely close to this

Section 2.2 I think that it is more natural to describe the coupling after section 2.1 and before the individual model setups (move 2.2 before section 2.1.1 and 2.1.2)

Line 125. Do you use CICE_Finalmod.F90?

Line 230 I think that ice sheet is normally used for glaciers. I would rephrase to ice cover

Line 330 I would replace roughly one month with the date it was reinitialized.

Line 376 reference figure 1 and the trajectories

Figure 4a I miss an explanation why the RMS error increases in wintertime.

Section 3.2.1/ Figure 5

I am not sure whether this adds value when figure 6 is included. The ocean is not in focus but the balance between 3 large figures and ~10 lines of text seems a bit off

Section 3.2.2

Please check references. I think that there are some, which do not match after removing a figure.

Line 394 – 398 some more details about the distribution would be nice.

Line 446 – 451 Comment There is a contradiction in running nested models. You would like to resolve the physics better and based on this get a different result. On the other hand, you also want the model to be similar on the boundary in order not to create strange behaviors there.

Line 449 I would remove matching

Line 460 and 462 hallo -> halo

---

## Author Response (AR2)

Dear Editor,

To begin with, we wish to thank the last critics from the Anonymous Referee #2, which we considered in revising the manuscript. For the sake of clarity, we reproduce the comments from the referee and then we present our answers. We indicate the lines of the paper where changes listed below took place, when relevant, using as reference the revised manuscript without the tracked changes.
All implemented changes may be tracked down in the attached files with the revised manuscript.

**Reviewer comments and our answers**

This is the second review of the manuscript that demonstrates the value of boundary conditions that includes dynamics. The manuscript has improved however I still request some minor changes. One is the missing conclusion.

**Answer:** A Conclusion was added to the manuscript (lines 484-496).

Abstract line 29. I assume that the improvement is due to the use of A4 oceanic boundary conditions rather than the nested area. If this the case it should be mentioned here.

**Answer:** Mentioned as suggested (line 30).

Line 51 and 52
"Also, knowledge about the possibility of ice in an area might be more important for applications than the specific details of the sea ice cover."
I am sure what is referred to here. Is it that the results are used as statistics of a hindcast for planning or is it that an approximate sea ice cover is good enough for some application? Please clarify

**Answer:** The main idea here is that when it comes to operational forecast applications, such as navigation, it may be more important to be sure about the presence/absence of sea ice in a given area that to have an accurate prognostic of sea ice properties. We added a small sentence to specify "navigation" (line 53) as an example of an application, hoping this will make the sentence clearer.

Line 68: CICE do include two packages from v6 and onwards, however you models are version 5.1.2 and something close to v6.0.0 (I assume). I am not sure whether it add value to mention it or if it confuses more that you mention it.

**Answer:** We hope the sentence is clear enough since we write: **"**It includes two independent packages: CICE and Icepack. Sea ice dynamics ... **Previous versions did not have such a separation, but the code evolved over the last years towards a clear distinction between processes which are mainly horizontal and those that are mainly vertical/columnar (since CICE6)"** (lines 70-73).

Line 77 primary model for forecasting **of** sea ice conditions….

**Answer:** Done as suggested (line 100).

Line 82 should this refer to section 2.2?

**Answer:** Yes. Corrected in the revised manuscript (line 105). However, due to other suggestion (below) we moved the section describing the coupling and now it is section 2.2.1.

Line 86 could you add a reference to Arome-Arctic

**Answer:** We added a link to the web page of the The Norwegian Meteorological Institute where the model is referenced, a citation (lines 109-110) and the respective reference (lines 606-608).

Line 105 Here you could reference the cice 6.0.0alpha CICE Release Table · CICE-Consortium/CICE Wiki · GitHub as the code you got is likely close to this

**Answer:** We understand the importance of referencing a documented version. However, we believe that the version we use is a bit older than the one suggested by the referee. Therefore, we described it using the same words we used in a previous paper co-authored by Elizabeth Hunke and following her advice (that is one of the reasons why we have the impression that the version we used does not match exactly one of those versions listed in the Wiki):
Duarte, P., et al. (2017), Sea ice thermohaline dynamics and biogeochemistry in the Arctic Ocean: Empirical and model results, J. Geophys. Res. Biogeosci., 122, 1632–1654, doi:10.1002/2016JG003660
However, we added a few words and now the sentence reads as:
"CICE [with a "column package" for thermodynamics and biogeochemical processes developed as part of the Accelerated Climate Model for Energy (ACME) project, **close to CICE6.0.0 alpha (https://github.com/CICE-Consortium/CICE/wiki/CICE-Release-Table)**]"
However, and as we emphasize in the last paragraph of the Discussion: "the compatibility of the changes described in this study with the most recent versions of the 474 Los Alamos Sea Ice Model (CICE + ICEPACK, https://github.com/CICE-Consortium), since the files changed and listed in 475 Table 1 are similar to those of the most recent versions."

Section 2.2 I think that it is more natural to describe the coupling after section 2.1 and before the individual model setups (move 2.2 before section 2.1.1 and 2.1.2)

**Answer:** Thanks for the suggestion. Done as suggested. Now the coupling is described in 2.1.1 and former sections 2.1.1 and 2.1.2 became 2.1.2 and 2.1.3, respectively. We also corrected references in the text to the various sections.

Line 125. Do you use CICE_Finalmod.F90?

**Answer:** Yes. This was now added to the manuscript. The sentence became: "ROMS is the controlling software acting through the CICE drivers CICE_InitMod.F90, CICE_RunMod.F90 and CICE_FinalMod.F90 to initialize, run and finalize CICE" (line 82).

Line 230 I think that ice sheet is normally used for glaciers. I would rephrase to ice cover

**Answer:** Corrected as suggested. The sentence now is: "This means that the entire sea ice covered area inside the domain of the model…" (line 233).

Line 330 I would replace roughly one month with the date it was reinitialized.

**Answer:** Done as suggested (lines 335-336 ).

Line 376 reference figure 1 and the trajectories

**Answer:** Since here we write about the observations used to evaluate model results, we have chosen to refer instead 2.3.2 (line 382) section where all details regarding these observations are given, both those from the N-ICE2015 expedition (referencing Figure 1) and those from satellite data. Please note that the reallocation of the description of model coupling implied changes in the numbering of several sections.

Figure 4a I miss an explanation why the RMS error increases in wintertime.

**Answer:** This was already answered in our previous response. We reproduce here what we wrote then: "We suspect that the increase in RMSE towards wintertime is related to Barents-2.5 km known cold sea surface temperature bias manifesting itself more in the winter."

Section 3.2.1/ Figure 5
I am not sure whether this adds value when figure 6 is included. The ocean is not in focus but the balance between 3 large figures and ~10 lines of text seems a bit off

**Answer:** We would rather keep this figure to illustrate vertical details about ocean model biases that show that, at least in comparison with TOPAZ4 results, the S4K ocean component is doing relatively well. We agree that the text associated with these figures is rather small, but it synthesizes the main messages about them.

Section 3.2.2
Please check references. I think that there are some, which do not match after removing a figure.

**Answer:** These references to the various figures were corrected, mostly by removing the wrong references to Fig. 10.

Line 394 – 398 some more details about the distribution would be nice.

**Answer:** Done as suggested – more details added (lines 400-406).

Line 446 – 451 Comment There is a contradiction in running nested models. You would like to resolve the physics better and based on this get a different result. On the other hand, you also want the model to be similar on the boundary in order not to create strange behaviors there.

**Answer:** Yes, we fully agree.

Line 449 I would remove matching

**Answer:** Removed as suggested.

Line 460 and 462 hallo -> halo

**Answer:** Corrected as suggested.

13 May 2022, Pedro Duarte (on behalf of all co-authors)

---

## Author Response (AR3)

Dear Editor,

Thank you and Polina Shvedko for letting us know about the color problems of some of our figures. We changed Figures 4, 5 and 7 to a different color scheme that should be visible to readers with color vision deficiencies. We also changed one of the vertical lines in Figure 4a to a dashed type to make sure that it is easily distinguishable from the vertical red line for all readers.

With my best regards,

Pedro Duarte (on behalf of all co-authors)